# Is the Earth's Magnetic Field a Constant? A Legacy of Poisson

**Jean-Louis Le Mouël [1], Fernando Lopes [1,\*], Vincent Courtillot [1], Dominique Gibert [2] and Jean-Baptiste Boulé [3]**

[1] Institut de Physique du Globe de Paris, Université de Paris, 75005 Paris, France; lemouel@ipgp.fr (J.-L.L.M.); courtil@ipgp.fr (V.C.)

[2] Laboratoire de Géologie de Lyon, Terre, Planètes et Environnement, Université de Lyon 1, ENSL, CNRS, UMR 5276, 69622 Lyon, France; dominique.gibert@univ-lyon1.fr

[3] CNRS UMR7196, INSERM U1154, Museum National d'Histoire Naturelle, 75005 Paris, France; jean-baptiste.boule@mnhn.fr

\* Correspondence: lopesf@ipgp.fr

**Abstract:** In the report he submitted to the Académie des Sciences, Poisson imagined a set of concentric spheres at the origin of Earth's magnetic field. It may come as a surprise to many that Poisson as well as Gauss both considered the magnetic field to be constant. We propose in this study to test this surprising assertion for the first time, evoked by Poisson in 1826. First, we present a development of Maxwell's equations in the framework of a static electric field and a static magnetic field in order to draw the necessary consequences for the Poisson hypothesis. In a second step, we see if the observations can be in agreement with Poisson. To do so, we choose to compare (1) the polar motion drift and the secular variation of Earth's magnetic field, (2) the seasonal pseudo-cycles of day length together with those of the sea level recorded by different tide gauges around the globe and those of Earth's magnetic field recorded in different magnetic observatories. We then propose a mechanism, in the spirit of Poisson, to explain the presence of the 11-year cycle in the magnetic field. We test this mechanism with observations, and finally, we study closely the evolution of the $g_{1,0}$ coefficient of the International Geomagnetic Reference Field (IGRF) over time.

**Keywords:** Poisson theory; Earth's magnetic field; constant intensity; length of day; polar motion





## 1. Introduction

The birth of geomagnetism as a science can be dated to 8 August 1269 on that day, Petrus Peregrinus wrote three letters ([1]; see f.i. Courtillot and Le Mouël [2]) during the siege of the city of Lucera in the Italian region of Puglie. The letters can be considered the first scientific article on geomagnetism, 331 years before the famous *De Magnete* by Gilbert ([3]). Expressed in modern terms, Peregrinus wrote that Earth has a magnetic field with a dipolar structure and that some rocks or minerals are magnetized. One could carve a sphere out of magnetite, pierce a hole through its center, and it would oscillate around the northward direction, a thought experiment that announced the compass.

In the three centuries that unfurled since then, a lot has been discovered: the variation of inclination with latitude, the daily, annual and other periodic variations ([4–6]), irregular variations, such as events linked to solar activity, the secular variation and its sudden jerks ([7–9]). In order to handle an ever increasing data base, magnetic indices (e.g., aa, Dst, Kp, etc.) were introduced (e.g., [4,6,10–17]). It was found that quasi periodical variations of the magnetic field ([4–6]), and also sun spots ([18]) followed a [19] power law with exponent $-5/3$.

In a series of papers that started with Le Mouël in 1984 ([20]) and continued with Jault et al. ([21]), and Jault and Le Mouël ([22–24]), these authors found that the trends of (1) magnetic secular variation, (2) polar motion and (3) length of day (lod) are strongly correlated. They proposed to explain these observations with a coupling mechanism, in

which flow in a cylinder tangent to the core and the rotation axis exchange torques at the core–mantle boundary. The solution for flow on the cylinder (see [24], system 6) is the same as that generated by an internal gravitational wave in a rotating fluid (e.g., [25]) also known as Proudman ([26]) flow. But this mechanism encounters serious difficulties with the orders of magnitude of physical parameters, such as the core–mantle boundary (CMB) topography. Also, the torque exerted by the fluid pressure and the electromagnetic torque are too weak to validate the model as concluded by Jault and Le Mouël ([24]).

Actually, the model of a cylinder tangent to the core is very close in spirit to that envisioned by Poisson ([27]). In the report he submitted to the Académie des Sciences, prior to an oral communication, Poisson imagined a set of concentric spheres in place of a cylinder. Poisson ([27]) was the first scientist to describe the magnetic field as a series of spherical harmonics, a decade before Gauss did ([28], Part 5, chapter 1, *Allgemeine Theorie des Erdmagnetismus*). He also invented a technique to measure the absolute value of the horizontal component of the magnetic field ([29,30]), seven years before Gauss ([31]) did.

It may come as a surprise to many that Poisson ([27]) as well as Gauss ([28]) both considered the magnetic field to be constant. This was an axiomatic basis for the development into spherical harmonics. There are very clear statements to this effect in the writings of both scientists. In the work of Poisson ([27]), page 49, one reads (our translation from the French): "*We will assume that the hollow sphere be magnetized under the influence of a force that be the same in magnitude and direction for all its points, such as the magnetic action of Earth, for instance*". And in page 54: "*Since time does not enter these formulae, a consequence is that, after the first instants of rotation, that we did not mention, the action of the rotating sphere on a given point will be constant in magnitude and in direction*". Gauss ([28]), part 5, chapter 1, paragraph 2, page 6, writes (our translation from the German): "*. . . magnetism consists only in galvanic currents (that is, constant currents) that persist in the smallest parts of the bodies . . .*". Gauss develops his theory very quickly (pages 18 to 23, paragraphs 14 to 27), without any physical proof. And his mathematical proof is exactly that found by Legendre ([32]) or Laplace ([33]) for the gravitational field. In contrast, the 130 pages of Poisson's memoir ([27]) are devoted both to the full physical and mathematical proofs of the magnetic field description.

Given that Poisson's work ([27]) precedes and is more complete than that of Gauss ([28]), it is only fair to recognize that the former was the first to develop the magnetic field in spherical harmonics and to state that this magnetic field was constant.

We do not propose to follow, as Poisson ([27]) did, the description of a magnetic field based on Maupertuis' ([34]) principle of least action but rather to pursue our previous presentation of the laws of gravitation of Lopes et al. ([35]) following Lagrange ([36]) and extend it to the case (and consequences) of the constancy of the electric and magnetic fields, giving their full physical meaning to the moments. This is the aim of Section 2. In Section 3, we try to explain the variations that one can measure within the frame of Poisson's paradigm ([27]). We confront the previous theoretical developments with modern observations in Section 4 and conclude in Section 5.

## 2. On the Constancy of the Magnetic Field

*2.1. Some Consequences of Maxwell's Equations*

A part of the demonstrations and results are known and detailed in various textbooks (see Landau and Lifshitz [37]).

One can learn a lot from the Lagrangian approach to the derivation of Maxwell's equations. Following Maupertuis ([34]), one only needs to know the action of a moving charged particle in an electromagnetic (EM) field, associated with the action of its interaction with that field, to derive the first pair of equations:

$$\mathrm{rot}\mathbf{E} = -\frac{1}{c}\frac{\partial \mathbf{H}}{\partial t} \tag{1a}$$

$$\mathrm{div}\mathbf{H} = 0 \tag{1b}$$

Adding the action of field EM, one obtains the second pair:

$$\text{rot}\mathbf{H} = \frac{1}{c}\frac{\partial \mathbf{E}}{\partial t} + \frac{4\pi}{c}\mathbf{j} \tag{1c}$$

$$\text{div}\mathbf{E} = 4\pi\rho \tag{1d}$$

Equations (1a)–(1d) link in a symmetrical way the magnetic field (**H**) to the electric field (**E**). $c$ is the light velocity, and $\rho$ is the charged particle associated with the current density **j**. It is important to note that, without knowing the action of **EM**, one has access to important properties: the space component, which is actually the field H, is conserved (1b). By analogy to Euler's (the continuity equation from fluid mechanics) equation, "magnetic charges" do not exist. Equation (1a) implies that as soon as **H** varies with time, a field **E** that is perpendicular (rot) and in quadrature with **H** is created (instantly, hence the term $1/c$). The second pair of equations is fully symmetrical. Equation (1d) implies that there exists an "electric charge" ($\rho$) that locally deforms the field **E**. In a vacuum, (1d) has exactly the same physical meaning as (1b). But one must add a current density to the time variation of **E** to propagate the field **H** (1c). A side note on (1c), it is also known as the Maxwell–Ampere equation. The classical understanding is that magnetic fields can be generated in two different ways, either by electrical currents (Ampere's theorem), or by time changes of field **E**, or the sum of both. The Lagrangian approach clarifies the picture. An EM field is defined by its 4-vector potential $A_i(= \varphi, \mathbf{A})$, where $\varphi$ is the time component (called the scalar potential, linked to **E**) and **A** is the space component (called the vector potential and linked to **H**). Charges that move in the magnetic field must obey the same decomposition; one, therefore, introduces a 4-vector current density $j^i(= c\rho, \mathbf{j})$, with a scalar charge density ($\rho$) found in (1d) and a vector current density (**j**).

The EM field described by the Maxwell equations must be in one of the three following forms: electrostatic, magnetostatic or a propagating field (wave propagation). This will not be discussed further in this paper.

The tradition is firmly established in geomagnetism to use electromagnetic units in the CGS system. A real advantage is that in vacuum and approximately in air, the magnitudes of the vectors **B** and **H**, which are only truly distinct in magnetized material, are measured by the same number.

### 2.2. The Electrostatic Field

Equations (1a) and (1d) reduce to

$$\text{div}\mathbf{E} = 4\pi\rho \tag{2a}$$

$$\text{rot}\mathbf{E} = 0 \tag{2b}$$

**E** derives from scalar potential ($\varphi$):

$$\mathbf{E} = -\text{grad}\,\varphi$$

leading to the Poisson equation:

$$\text{div}(\text{grad})\,\varphi = \Delta\varphi = -4\pi\rho \tag{2c}$$

In a vacuum ($\rho = 0$), the scalar potential verifies the Laplace equation:

$$\Delta\,\varphi = 0$$

The field produced by a point charge ($e$) is directed along the vector, having the charge as one of its extremities. **E** is a radial field. The absolute value of **E** depends only on the distance $R$ to $e$. Applying the divergence theorem to (2a),

$$\text{div}\mathbf{E} \longmapsto \iiint_V \text{div}\mathbf{E}\, dV = \iint_S \mathbf{E}.d\mathbf{S}$$

The flux of **E** across a spherical surface with radius $R$ centered on $e$ is $4\pi R^2 \mathbf{E}$ and also equals $4\pi e$ from Gauss's theorem,

$\oint \mathbf{E}d\mathbf{f} = 4\pi \int edV$, $df$ being the surface integration element.

Finally, in vector form,

$$\mathbf{E} = \frac{e\mathbf{R}}{R^3} \qquad (2d)$$

Thus, the field produced by a point charge is inversely proportional to the square of the distance to $e$ (Coulomb's law). The potential associated with this field is

$$\varphi = \frac{e}{R} \qquad (2e)$$

and for a system of charges,

$$\varphi = \sum_i \frac{e_i}{R_i} \qquad (2f)$$

Let us observe this field (2f) at a distance that is large compared to the charge system's dimension and that is far enough so that their relative motions can be considered constant (in the sense of Lagrangian mechanics). This allows one to use the concept of "moment".

Let us choose a coordinate system whose origin lies within the charge system and let $r_i$ be their respective vector radii. The total observed potential at point $R_o$ is

$$\varphi = \sum \frac{e_i}{|\mathbf{R}_o - \mathbf{r}_i|} \qquad (3a)$$

For $R_o \gg r_i$ and thanks to the generalized Maclaurin expansion given by [32], we can develop (3a) in a series of powers of $\dfrac{r_i}{R_o}$, using the first-order formula:

$$f(\mathbf{R}_o - \mathbf{r}) \approx f(\mathbf{R}_o) - \mathbf{r}\,\text{grad}f(\mathbf{R}_o)$$

thus, (3a) becomes

$$\varphi = \frac{\sum e_i}{\mathbf{R}_o} - \sum e_i\mathbf{r}_i\text{grad}\frac{1}{\mathbf{R}_o} \qquad (3b)$$

The sum $\mathbf{d} = \sum e_i\mathbf{r}_i$ is the dipolar moment of the charge system. By analogy with a system of masses, this dipolar moment is the mathematical equivalent to the tensor of moments of inertia of order 2 in geodesy. If the sum of charges is zero, the dipolar moment does not depend on the choice of the origin ($\mathbf{r}' = \mathbf{r} + \mathbf{k}$), $\mathbf{k}$ constant, $\sum e_i = 0$, then $\mathbf{d}' = \sum e_i\mathbf{r}_i' = \sum e_i\mathbf{r}_i + \mathbf{k}\sum e_i = \mathbf{d}$).

The potential at large distances can be written as

$$\varphi = -\mathbf{d}\nabla\frac{1}{R_o} = \frac{\mathbf{d}\mathbf{R}_o}{R_o^3} \qquad (3c)$$

and

$$\begin{aligned} \mathbf{E} &= -\text{grad}\varphi \quad \left(= (\mathbf{d}\nabla)\nabla\frac{1}{R_o}\right)\\ &= -\text{grad}\frac{\mathbf{d}\mathbf{R}_o}{R_o^3} = -\frac{1}{R_o^3}\text{grad}(\mathbf{d}\mathbf{R}_o) - (\mathbf{d}\mathbf{R}_o)\text{grad}\frac{1}{R_o^3}\\ &= -\frac{\mathbf{d}}{R_o^3}\nabla\mathbf{R}_o - \frac{\mathbf{R}_o}{R_o^3}\nabla\mathbf{d} - \mathbf{d}\mathbf{R}_o\nabla\frac{1}{R_o^3}\\ &= -\frac{\mathbf{d}}{R_o^3} - 0 + \frac{3(\mathbf{n}\mathbf{d})\mathbf{n}}{R_o^3}\\ &= \frac{3(\mathbf{n}\mathbf{d})\mathbf{n} - \mathbf{d}}{R_o^3} \end{aligned}$$

where **n** is the unit vector oriented towards $R_o$. At large distances, the potential is inversely proportional to the square of distance and **E** to its cube. **E** is axially symmetrical about **d**. In a plane where the direction of **d** is that of the $z$ axis, the Cartesian components of **E** are

$$E_z = d. \frac{3\cos^2\theta - 1}{R_o^3}, \quad E_x = d. \frac{3\sin\theta\cos\theta}{R_o^3} \tag{3d}$$

and the radial and tangential components are:

$$E_r = d. \frac{2\cos\theta}{R_o^3}, \quad E_\theta = -d. \frac{\sin\theta}{R_o^3} \tag{3e}$$

$\theta$ being the angle between $z$ and $\mathbf{R}_o$. We note that Equations (3d) and (3e) are the same as those of the components of a dipolar magnetic field, with $d$ being replaced by $\frac{\mu_0}{4\pi}$ (e.g., LeMouel in [38], chapter 26, page 40, system 24).

As in [33], one can always develop the scalar potential $\varphi$ as a sum of contributions in ascending powers of $\frac{1}{R_0}$, the term $\varphi^{(n)}$ being proportional to $\frac{1}{R_o^{n+1}}$:

$$\varphi = \varphi^{(0)} + \varphi^{(1)} + \varphi^{(2)} + \dots \tag{4a}$$

The first term $\varphi^{(0)}$ is determined by the sum of all charges or masses for Laplace ([33]), for whom this term can never be zero. As we saw, when the sum of electric charges is zero, one is led to the electrostatic components (3d) and (3e). The second term, $\varphi^{(1)}$ is the dipolar one, determined by its dipolar moment d. One can continue the development in a Legendre ([32]) series. The next term would be

$$\varphi^{(2)} = \frac{1}{2} \sum e x_i x_j \frac{\partial^2}{\partial X_i \partial X_j} \frac{1}{R_o} \tag{4b}$$

where the sum is extended to all charges and where the $x$ coordinates are the components of **r** and the $X$ coordinates are those of $\mathbf{R}_o$. We note that

$$\Delta \frac{1}{R_o} \equiv \delta_{ij} \frac{\partial^2}{\partial X_i \partial X_j} \frac{1}{R_o} = 0$$

One can then write (4b) as

$$\varphi^{(2)} = \frac{1}{2} \sum e (x_i x_j - \frac{1}{3} r^2 \delta_{ij}) \frac{\partial^2}{\partial X_i \partial X_j} \frac{1}{R_o}$$

The tensor $D_{ij} = \sum e(3x_i x_j - r^2 \delta_{ij})$ is the quadrupolar moment of the charge system. Thus,

$$\varphi^{(2)} = \frac{D_{ij}}{6} \frac{\partial^2}{\partial X_i \partial X_j} \frac{1}{R_o} \tag{4c}$$

or, as done for the dipolar term in (3c),

$$\frac{\partial^2}{\partial X_i \partial X_j} \frac{1}{R_o} = \frac{X_i X_j}{R_o^5} - \frac{\delta_{ij}}{R_o^3}$$

and since $\delta_{ij} D_{ij} = D_{ii} = 0$, we have

$$\phi^{(2)} = \frac{D_{ij} n_i n_j}{2R_o^3} \tag{4d}$$

$n_i$ and $n_j$ are the unit vectors starting from $\mathbf{R}_o$ and oriented along the two axes of the quadrupolar moment. The eigenvalues of the tensor are such that $D_{ii} = 0$ and are, therefore, linked by

$$D_{xx} = D_{yy} = -\frac{1}{2}D_{zz}$$

If we write $D$ for component $D_{zz}$, the quadrupolar potential becomes

$$\varphi^{(2)} = \frac{D}{4\pi R_o^3}(3\cos^2\theta - 1) = \frac{D}{2R_o^3}\mathcal{P}_2(\cos\theta) \tag{4e}$$

in which one introduces the $\mathcal{P}_2$ Legendre ([32]) polynomial. One can generalize this construction; the $l$-order term is determined by a tensor of order $l$, the $2^l$-polar moment.

The mathematical development above is independent of the physical problem (gravity, magnetism, and electricity) one is interested in. It serves to illustrate the fundamental result from Legendre ([32]) regarding the theory of the gravitational attraction of masses as generalized by Laplace ([33]): as far as one observes from the far enough sources, the inverse of distance ($\frac{1}{\mathbf{R}_o - \mathbf{r}}$) is given by

$$\frac{1}{|\mathbf{R}_o - \mathbf{r}|} = \frac{1}{\sqrt{\mathbf{R}_o^2 + \mathbf{r}^2 - 2\mathbf{r}\mathbf{R}_o\cos\chi}} = \sum_{l=0}^{\infty} \frac{\mathbf{r}^l}{\mathbf{R}_o^{l+1}}\mathcal{P}_l(\cos\chi) \tag{5a}$$

Let us introduce the pairs of spherical angles $\Theta$, $\Phi$ and $\theta$, $\varphi$ formed respectively by vectors and with the given coordinate axes, and apply the addition theorem of spherical functions:

$$\mathcal{P}_l(\cos\chi) = \sum_{m=-l}^{l} \frac{(l-|m|)!}{(l+|m|)!}\mathcal{P}_l^{|m|}(\cos\Theta)\mathcal{P}_l^{|m|}(\cos\theta)e^{-im(\Phi-\varphi)} \tag{5b}$$

where $\mathcal{P}_l^m$ are associated Legendre polynomials. Let us also introduce the spherical functions:

$$\mathcal{Y}_{lm}(\theta, \varphi) = (-1)^m i^i \sqrt{\frac{(2l+1)(l-m)!}{4\pi(l+m)!}}\mathcal{P}_l^m(\cos\theta)e^{im\varphi}, \quad m \geq 0. \tag{5c}$$

Integrating (5b) and (5c) in (5a), one finally obtains the expression for the inverse of a distance on the sphere:

$$\frac{1}{|\mathbf{R}_o - \mathbf{r}|} = \sum_{l=0}^{\infty}\sum_{m=-l}^{l} \frac{r^l}{R_o^{l+1}}\frac{4\pi}{2l+1}\mathcal{Y}_{lm}^*(\Theta, \Phi)\mathcal{Y}_{lm}(\theta, \varphi) \tag{5d}$$

Developing each term in Equation (3a), one obtains the expression for the term of order $l$ of the potential:

$$\varphi^{(l)} = \frac{1}{R_o^{l+1}} \sum_{m=-l}^{l} \sqrt{\frac{4\pi}{2l+1}}\mathcal{Q}_m^{(l)}\mathcal{Y}_{lm}^*(\Theta, \Phi) \tag{5e}$$

where the $2l + 1$ quantities $\mathcal{Q}_m^l$ constitute the $2^l$-polar moment of the system of charges, defined by

$$\mathcal{Q}_m^l = \sum_i e_i r_i^l \sqrt{\frac{4\pi}{2l+1}}\mathcal{Y}_{lm}(\theta_i, \varphi_i) \tag{5f}$$

At this point, let us underline why we have taken the trouble to recall this (at least in large part) classical derivation, which is likely taught in all graduate and even undergraduate physics programs. In this section, we saw that in the case of a static field $\mathbf{E}$, Coulomb's

law (2e) imposes itself, and the field is radial. In the case of a system of charges, if one is too close to the system, the interactions of the charges *forbid one to use the Lagrangian concept of moment*. Thus, one must remain *far from the system*. But as found by Legendre ([32]) and Laplace ([33]), when attempting to define the shape of the attraction field of masses, it is seen that the **E** field involves the same constraints, i.e., is *electrostatic*. It is only because we are with a static field that the notion of the *inverse of a distance* takes its full meaning and that we can develop it into *spherical harmonics*.

We can now undertake the same analysis in the case of the magnetic field.

### 2.3. The Magnetostatic Field

As we have seen above, Maxwell's Equations (1b) and (1c) imply that the magnetic field **H**, created by charges in finite motion, remaining in a finite region of space (1b), whose impulses always retain finite values, has a stationary character that we wish to analyze further.

The two equations are now

$$\text{div } \mathbf{H} = 0 \tag{6a}$$

$$\text{rot } \mathbf{H} = \frac{4\pi}{c}\mathbf{j} \tag{6b}$$

The vector potential **A** associated with **H** is defined by rot **A** = **H**. Carried into (6b), it becomes

$$\text{grad div}\mathbf{A} - \Delta\mathbf{A} = \frac{4\pi}{c}\mathbf{j}$$

**A** being defined in a non-unequivocal way, one can impose an arbitrary condition, such as div **A** = 0. The previous line then becomes the Poisson equation:

$$\Delta\mathbf{A} = -\frac{4\pi}{c}\mathbf{j} \tag{6c}$$

(6c) is analogous to (2c), charge density $\rho$ being replaced by current density $\frac{\mathbf{j}}{c}$. By analogy with the electrical potential, we can write

$$\mathbf{A} = \frac{1}{c}\int \frac{\mathbf{j}}{R}dV \tag{6d}$$

where $R$ is the distance from the observation point to the volume element $dV$. The integral in (6d) can be replaced by a sum and the current density by $\rho\mathbf{v}$. By analogy with the scalar potential (3a), the vector potential becomes

$$\mathbf{A} = \frac{1}{c}\sum \frac{e_i\mathbf{v}_i}{R_i} \tag{6e}$$

This is how charges are introduced in a vector linked to the magnetic field, without risking the mistake of writing that the magnetic field derives from a scalar potential.

As done above for the electrostatic field, one can calculate the effect of the moving charges in a reference system with its origin within the charge distribution, with the same notations for vectors $\mathbf{r}_i$ and distance $\mathbf{R}_o$. (6e) becomes

$$\mathbf{A} = \frac{1}{c}\sum \frac{e_i\mathbf{v}_i}{|\mathbf{R}_o - \mathbf{r}_i|} \tag{7a}$$

The main difference between the scalar and vector potentials is that for **E**, one only has the effect of fixed charges or motion as a rigid block, whereas for **H**, what counts is the uniform velocity of charges imposed by (1b). This is the reason why Poisson [27]'s title is *Du magnétisme en mouvement* (of magnetism in motion). And this is the main reason why one cannot write a physical description of a magnetic field as a series of spherical harmonics.

However, some remarkable consequences can be derived. For instance, one can always write in a development as a Legendre ([32]) series analogous to (3b), to the first order:

$$\mathbf{A} = \frac{1}{cR_o} \sum e\mathbf{v} - \frac{1}{c} \sum e\mathbf{v}(\mathbf{r}\nabla \frac{1}{R_o})$$

One can write $\sum e\mathbf{v} = \frac{d}{dt} \sum e\mathbf{r}$, but the mean value of the derivative varying in a finite interval is 0.

$$\mathbf{A} = -\frac{1}{c} \sum e\mathbf{v}(\mathbf{r}\nabla \frac{1}{R_o}) = \frac{1}{cR_o^3} \sum e\mathbf{v}(\mathbf{r}\mathbf{R}_o) \tag{7b}$$

Note that $\mathbf{v} = \dot{\mathbf{r}}$ and since $\mathbf{R}_o$ is a constant vector,

$$\sum e(\mathbf{R}_o\mathbf{r})\mathbf{v} = \frac{1}{2}\frac{d}{dt} \sum e\mathbf{r}(\mathbf{R}_o\mathbf{r}) + \frac{1}{2} \sum e[\mathbf{v}(\mathbf{r}\mathbf{R}_o) - \mathbf{r}(\mathbf{v}\mathbf{R}_o)]$$

Carrying this expression in (7b), the mean value of the first term is again 0, thus,

$$\mathbf{A} = \frac{1}{2cR_o^3} \sum e[\mathbf{v}(\mathbf{r}\mathbf{R}_o) - \mathbf{r}(\mathbf{v}\mathbf{R}_o)] \tag{7c}$$

One recognizes a vector product in (7c). Let us introduce the magnetic moment $\mathbf{m} = \frac{1}{2c} \sum e\mathbf{r} \times \mathbf{v}$. Equation (7c) becomes

$$\mathbf{A} = \nabla\frac{1}{R_o} \times \mathbf{m} = \frac{\mathbf{m} \times \mathbf{R}_o}{R_o^3} \tag{7d}$$

Whereas $\frac{1}{R_o}$ verifies Laplace's equation, its modification by the rotational of $\mathbf{m} = \frac{1}{2c} \sum e\mathbf{r} \times \mathbf{v}$ does not. With the expression for the vector potential (7d), one can derive the magnetic field. Given the formula rot $\mathbf{a} \times \mathbf{b} = (\mathbf{b}\nabla)\,\mathbf{a} - (\mathbf{a}\nabla)\,\mathbf{b} + \mathbf{a}\,\text{div}\,\mathbf{b} - \mathbf{b}\,\text{div}\,\mathbf{a}$, one finds

$$\mathbf{H} = \text{rot}\,\mathbf{m} \times \frac{\mathbf{R}_o}{R_o^3} = \mathbf{m}\,\text{div}\frac{\mathbf{R}_o}{R_o^3} - (\mathbf{m}\nabla)\frac{\mathbf{R}_o}{R_o^3}$$

Since with $\mathbf{R}_o \neq 0$,

$$\text{div}\frac{\mathbf{R}_o}{R_o^3} = \mathbf{R}_o\text{grad}\frac{1}{R_o^3} + \frac{1}{R_o^3}\text{div}\mathbf{R}_o = 0,$$

and

$$(\mathbf{m}\nabla)\frac{\mathbf{R}_o}{R_o^3} = \frac{1}{R_o^3}(\mathbf{m}\nabla)\mathbf{R}_o + \mathbf{R}_o(\mathbf{m}\nabla\frac{1}{R_o^3}) = \frac{\mathbf{m}}{R_o^3} - \frac{3\mathbf{R}_o(\mathbf{m}\mathbf{R}_o)}{R_o^5},$$

then

$$\mathbf{H} = \frac{3\mathbf{n}(\mathbf{m}\mathbf{n}) - \mathbf{m}}{R_o^3}, \tag{7e}$$

where $\mathbf{n}$ is the unit vector along direction $\mathbf{R}_o$. If the ratio of mass to charge is the same for all charges in the system, then

$$\mathbf{m} = \frac{1}{2c} \sum e\mathbf{r} \times \mathbf{v} = \frac{e}{2mc} \sum m\mathbf{r} \times \mathbf{v}$$

Finally, if all velocities of all charges are such that $v \ll c$, then $m\mathbf{v}$ is the impulsion $\mathbf{p}$ of the charge and

$$\mathbf{m} = \frac{e}{2mc} \sum \mathbf{r} \times \mathbf{p} = \frac{e}{2mc}\mathcal{M} \tag{8a}$$

where $\mathcal{M} = \sum \mathbf{r} \times \mathbf{p}$ is the angular momentum of the system. Here, the ratio of the magnetic moment to the angular momentum is a constant.

Let us now consider a system of charges placed in a uniform and constant, external magnetic field **H**. The (time-averaged) force exerted on the system (by the field **H**) over time is 0. Indeed, according to the Lorentz force ([39]) $\mathcal{F} = \sum \frac{e}{c} \mathbf{v} \times \mathbf{H} = \frac{d}{dt} \sum \frac{e}{c} \mathbf{r} \times \mathbf{H}$, $\mathcal{F}$ is the temporal derivative (taken between two finite times) of a quantity involving **H**. It is known as Maxwell's 5th equation. On the other hand, the time average of the moment of forces is $\mathcal{K} = \sum \frac{e}{c} \mathbf{r} \times (\mathbf{v} \times \mathbf{H})$ *which is not 0*. Writing explicitly the double vector product,

$$\mathcal{K} = \sum \frac{e}{c} \{ \mathbf{v}(\mathbf{r}\mathbf{H}) - \mathbf{H}(\mathbf{v}\mathbf{r}) \} = \sum \frac{e}{c} \{ \mathbf{v}(\mathbf{r}\mathbf{H}) - \frac{1}{2}\mathbf{H}\frac{d}{dt}\mathbf{r}^2 \}$$

one obtains simply

$$\mathcal{K} = \mathbf{m} \times \mathbf{H}$$

The Lagrangian of a system of charges placed in an external, constant and uniform magnetic field is

$$\mathcal{L}_H = \sum \frac{e}{c}\mathbf{A}\mathbf{v} = \sum \frac{e}{2c}(\mathbf{H} \times \mathbf{r})\mathbf{v} = \sum \frac{e}{2c}(\mathbf{r} \times \mathbf{v})\mathbf{H} \tag{8b}$$

Introducing the magnetic moment (8b) becomes simply,

$$\mathcal{L}_H = \mathbf{m}.\mathbf{H} \tag{8c}$$

By analogy, in a uniform electric field, the Lagrangian of a system with 0 total charge and a dipolar moment includes the term

$$\mathcal{L}_E = \mathbf{d}.\mathbf{E} \tag{8d}$$

Let us consider a system of charges undergoing a finite motion in a field **E** with central symmetry, due to a motionless particle. Let us shift from a motionless system of coordinates to a system undergoing a uniform rotation about an axis passing through the motionless particle. The velocity **v** of the particle in the new system is linked to its velocity **v** in the old system by $\mathbf{v}' = \mathbf{v} + \Omega \times \mathbf{r}$, where **r** is the particle's vector radius and $\Omega$ the angular velocity of the rotating coordinate system. In the fixed system, the Lagrangian of charges is

$$\mathcal{L} = \sum \frac{mv'^2}{2} - \mathcal{U},$$

where $\mathcal{U}$ is the potential energy of charges in the external field E. In the new system, the Lagrangian becomes

$$\mathcal{L} = \sum \frac{m}{2}(\mathbf{v} + \Omega \times \mathbf{r})^2 - \mathcal{U}.$$

If the ratio $\frac{e}{m}$ of charge to mass is the same for all particles, and if one writes

$$\Omega = \frac{e}{2mc}\mathbf{H}$$

then, for sufficiently small values of **H** such that one can neglect terms in $\mathbf{H}^2$, the Lagrangian takes the form

$$\mathcal{L} = \sum \frac{mv^2}{2} + \frac{1}{2c}\sum e(\mathbf{H} \times \mathbf{r})\mathbf{v} - \mathcal{U} \tag{8e}$$

It is remarkable that the two Lagrangians in (8b) and (8c) are the same. In summary, the Lagrangian of charges in finite motion in an electric field produced by a motionless rotating particle (8c) is the same as the Lagrangian of a system of charges placed in a constant and uniform magnetic field (8b). In slightly different, more readable terms, the behavior of a system of charges with the same $\frac{e}{m}$ ratio executing a finite motion in a field **E**

with central symmetry and a weak and uniform field **H** is equivalent to the behavior of the same charge system in field **E** with respect to a uniformly rotating coordinate system with angular velocity $\Omega$. This is Larmor's theorem (cf. [40], chapter VI, [41]).

Let us now evaluate the variation of the mean angular momentum $\mathcal{M}$. The variation of $\mathcal{M}$ is equal to the moment $\mathcal{K}$ of forces applied to the system. Thus, $\mathcal{K} = \mathbf{m} \times \mathbf{H} = \dfrac{d\mathcal{M}}{dt}$. If the ratio $\dfrac{e}{m}$ is the same for all the system's particles, then $\mathcal{M}$ is proportional to **m** (cf. (8a)), and

$$\frac{d\mathcal{M}}{dt} = -\Omega \times \mathcal{M}. \tag{8f}$$

Vector $\mathcal{M}$, and thus vector **m**, both rotate with angular velocity $-\Omega$ about the field direction; its absolute value and the angle it makes with respect to the field direction are constant.

### 3. Some Further Remarks on Section 2

Most, if not all, modern studies of the geomagnetic field have it varying in time, keeping its first-order dipolar configuration and 'ready' to be analyzed with spherical harmonics. But a time variable **E** or **H** field implies wave propagation, hence physics described by Helmholtz equations, not Legendre. Let us describe what happens with a variable field in the equations of Section 2.

The only link between the field geometry and the physics of the problem, that is, the $e$ charges and their positions in the reference system, is (5f), which we recall here:

$$\mathcal{Q}_m^l = \sum_i e_i r_i^l \sqrt{\frac{4\pi}{2l+1}} \mathcal{Y}_{lm}(\theta_i, \varphi_i)$$

One needs to visualize the shapes of spherical harmonics that are "physically" represented by the coefficients $\mathcal{Y}_{lm}$. Figure 1 displays the first four axial multipoles, that is, the eigenvectors $\mathcal{Y}_{1,0}$, $\mathcal{Y}_{2,0}$, $\mathcal{Y}_{3,0}$ and $\mathcal{Y}_{4,0}$ (from left to right and top to bottom) corresponding to the so-called Gauss coefficients $g_{1,0}$, $g_{2,0}$, $g_{3,0}$ and $g_{4,0}$. The eigenvectors are a constant of the problem, as long as this representation retains a physical meaning. For example, a dipolar magnetic field must always have its principal axis aligned with the axis going from the origin to the geographic North pole. In this paradigm, a variable field would involve only modifications of the space–time physics of term $e^i r_i^l$ in the sum (5f). But this constraint cannot be satisfied due to either of the following:

1.　The position term $r_i^l$ of each moving particle fluctuates with time so that the Legendre–Laplace condition (5a) is not satisfied any more, that is, the inverse distance $\dfrac{1}{|\mathbf{R}_o - \mathbf{r}|}$ is no more a natural solution of the Laplacian. One would need to introduce time but then the Laplacian would have to be replaced by a Dalembertian, i.e., a different problem.

2.　It is the number and/or quality of the charges that would change with time. But then the nature of the core would change with time, and one would need to find a physical mechanism that would explain how the field intensity could decrease (as is the case at present), yet could have increased and even reversed in the past.

Poisson's reasoning ([27]) is simpler. As early as page 7 of his memoir, above his first equation, he linked gravitation and magnetism. So did Gauss ([28]), and this was much developed by Heaviside ([42]). This was more than a simple physical analogy.

Recall that modern geophysicists use Talwani's algorithms ([43,44]) to determine the mass (respectively, dipolar) distributions based on microgravimetric (respectively, magnetic) measurements using the same equation. Poisson ([27]) writes: '*The differential volume element, corresponding to point M' [The magnetic volume], will have $h^3 d\chi d\xi d\eta$ as its expression; we will write $\mu' h^3 d\chi d\xi d\eta$ for the amount of free fluid it contains, $\mu'$ being positive or negative according to this fluid being boreal or austral. This coefficient will be a function of $\chi, \xi, \eta$*

*depending on the distribution of the two fluids inside the magnetic elements. If they are moving, μ'
will vary with time, but the total quantity of free fluid belonging to the same element must always
be zero, so we will always have*

$$\int \mu' d\chi d\xi d\eta = 0 \qquad (I)$$

*The integral extending to the entire volume of the magnetic element'.*

This short quotation is from the very beginning of Poisson's derivation ([27]); in more
modern terms, div **H** = 0. What enters a volume element and what leaves it is constant: the
field is stationary.

From Lagrange's standpoint ([36]), Poisson ([27]) implies that at the origin of the magnetic field is a rather undeformable object, a solid body consisting of a set of electromagnetic
point sources whose respective distances do not vary, or that can be considered as such
at the distance at which its effects are observed. We note that the tensor of a quadrupolar
electrical moment $D_{ij} = \sum e(3x_i x_j - r^2 \delta_{ij})$ is similar to the order 2 inertia moment tensor of
a rotating solid body $I_{ik} = \sum m(x_i^2 \delta_{ik} - x_i x_k)$.

One then concludes that the space–time variations of electric charges that would
not result in a constant field would make *it impossible to construct either an electric or a
magnetic moment.*

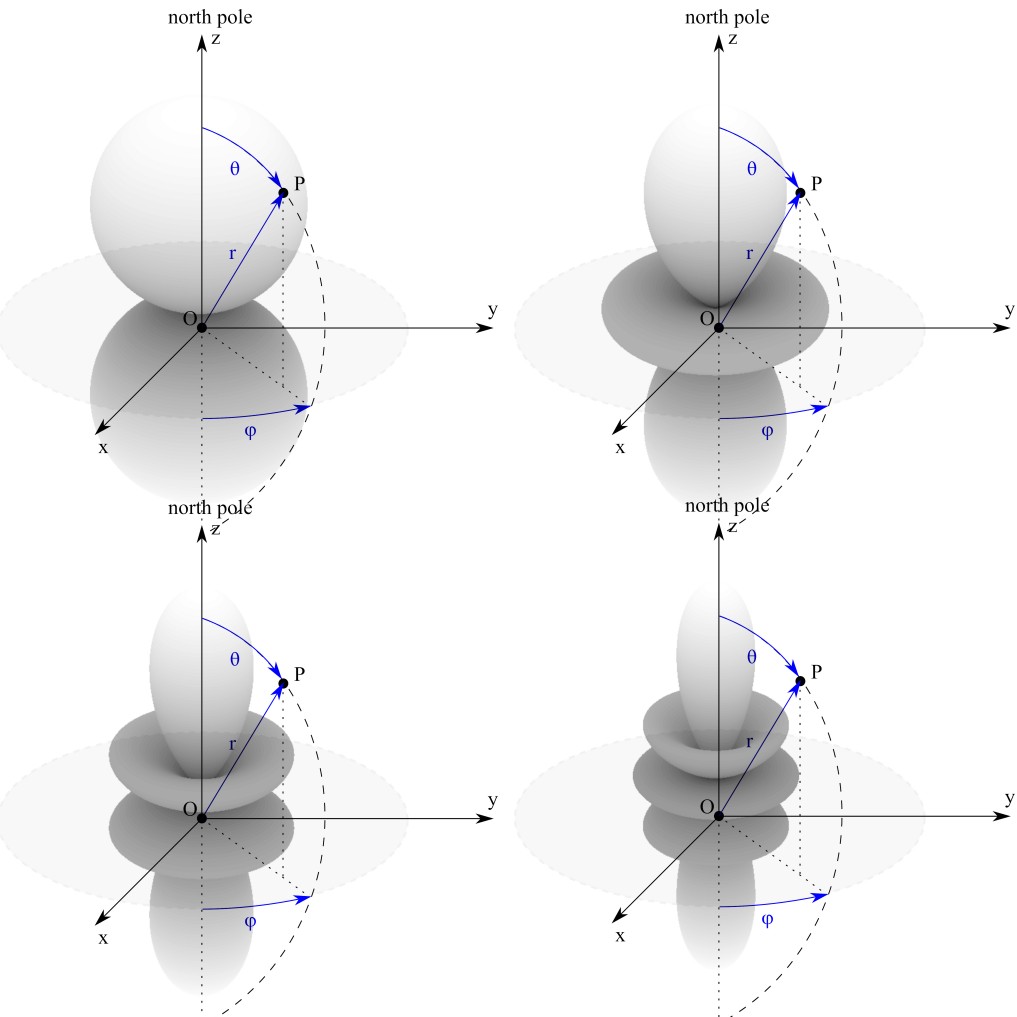

**Figure 1.** Spherical harmonics (from left to right and top to bottom) $\mathcal{Y}_{1,0}$, $\mathcal{Y}_{2,0}$, $\mathcal{Y}_{3,0}$ and $\mathcal{Y}_{4,0}$.

## 4. Reconciling Modern Observations with Poisson's Theory

### 4.1. On the Drift of the Magnetic Dipole

In Sections 2 and 3, we showed that the only way a magnetic field can be written as the sum of multipolar potentials is that (similar to masses composing a solid rotating body) charges generating the electric and magnetic fields should move uniformly in space and time. In other words, the field must be stationary. We also know that the field intensity fluctuates and that this secular variation is morphologically similar to the drift of the rotation pole, called the Markowitz (or Markowitz–Stoyko, [45,46]) drift. The annual oscillations of the field are morphologically similar to those of the length of the day ([21–24]).

One can reconcile Poisson ([27]) and observations by picturing a dipole that oscillates about the geographical North, this oscillation sharing the same excitation (forcings) that acts on the polar motion. Let us first illustrate this idea. In Figure 2, we show the eigenvector $\mathcal{Y}_{1,0}$ associated with the Gauss coefficient $g_{1,0}$. CLF stands for the Chambon-La-Forêt magnetic observatory. Point P is one of the elements of the magnetic volume of [27] composing the dipole. As seen in Section 2, the dipole action at CLF is characterized by the distance P-CLF on the sphere. If the dipole is tilted (Figure 2, right), this distance changes, and so do the coordinates of CLF in the two dipole reference systems (Figure 2, left and right).

Following Lagrange ([36]), a number of authors (e.g., [35,47–51]) have shown how and how much astronomical forces influence Earth's rotation, in the same way its own weight perturbs the rotation of a spinning top, through an exchange of angular moments. The same astronomical forces perturb the number of sunspots, i.e., solar activity (e.g., [52–55]). Since [33], we know that masses at the surfaces of planets can re-organize under the influence of stresses acting on the rotation pole. The Liouville–Euler system of linear differential equations of the first order runs this re-organization (cf. [56], Section 3 for more details). From Sections 2 and 3, moving charges may be considered as a moving fluid in rotation about the dipole's symmetry axis. Any large-scale fluid motions must correspond to a block rotation about the Earth's rotation axis, as shown by Laplace and Poincaré. There are no other possibilities of natural motion at the first order. Inside as well as at its surface, the pattern of fluid motion must be the same (cf. [33,57]; see [58,59] for some illustrations).

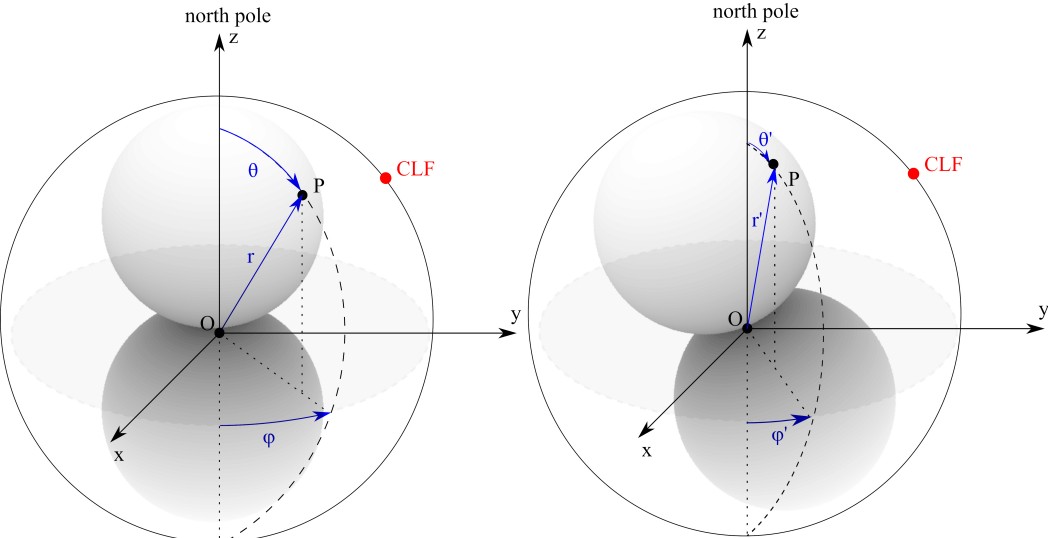

**Figure 2.** Eigenvector $\mathcal{Y}_{1,0}$ associated with Gauss coefficient $g_{1,0}$: to the left, the dipole is axial.

Possibly the longest series of magnetic measurements is that of declination D compiled for Paris (France) by Alexandrescu et al. ([60,61]). The series starts in 1541, but we select values from 1781 onward, when the sampling becomes more regular. The Brest sea level data start in 1807. The polar motion (see Appendix A for more details)data are defined as the modulus of the horizontal displacement of the pole in a plane tangential to Earth ($m_1 + i\, m_2$); it starts in 1846 and therefore sets the length of our analyses.

We applied singular spectrum analysis (SSA, e.g., [62]) in order to extract the trend (this sub-section) and the annual and semi-annual components (next subsection) from the three time series of the sea level at Brest, magnetic field at CLF and length of day. SSA uses the mathematical properties of descending order diagonal matrices (Hankel, Toeplitz matrices, e.g., [63]) and their orthogonalization by singular value decomposition (SVD, e.g., [64]).

In Figure 3a, we superimpose the SSA trends of declination in CLF in red, sea level at Brest in blue, and polar motion in black. We dealt with the sea-level series in Le Mouël et al. ([65]), in which we compared variations of the sea-level trends with the variations of Markowitz–Stoyko polar drift. We studied many aspects of polar motion in [49,66,67]. The time derivatives of the three trends are shown in Figure 3b. We also used the long dataset of Stepheson and Morrison ([68]) and Gross ([69]) for *lod* and compare its trend with the first derivative of magnetic declination (Figure 3c).

The trends of the three derivatives display very similar patterns. We showed before that this pattern is linked to the ephemerids of Uranus ([49,65]). The similarity between the derivative of the trend of the rotation pole and sea level is expected: as Earth shifts, its fluid envelope shifts as a solid in the same way. The two are almost in phase. For the magnetic field, there is a ∼20-year phase lag. It is in quadrature with polar motion around 1930, and catches up in the 1960s. This vindicates the mechanism advocated by Le Mouël et al. ([20,21]) and Jault et al. ([22–24]), that is, a transfer of moment at the core–mantle boundary. The phase lag would be due to the roughness of the CMB.

Laplace ([33]) showed theoretically that lod and pole motion are linked by a first-order derivative operator. We verified this with the observations of Lopes et al. ([67]). Figure 3c compares the (5 yr smoothed) mean value of the derivative of declination in Paris to the corresponding (5 yr smoothed) mean value of the length of day. The two series of mean values of solid Earth motions are in quadrature and show a ∼60 yr oscillation ([67], Figure 4). This 60 yr period is clearly present in the 2-hump pattern of lod as well as in the derivative of declination (blue curve, Figure 3c). As shown in Section 2, the angular and magnetic moments that are at the origin of the geomagnetic field are linked linearly (Equation (8c)). On the other hand, the angular momentum and the polar rotation axis are linked through a first-order time derivative (Equation (8a)). In the spirit of Larmor's theorem, we assume that the two phenomena (rotation and magnetism) are linked by the same external forcing; thus the former should be compared to the derivative of the latter. In the lower part of Figure 3c, the mean value of lod is offset by 60 years, leading to an almost perfect match of the patterns. We have not (yet) been able to explain this offset.

One cannot envision that the magnetic field (declination) variations would be due to intensity variations of components X, Y and Z since, as seen in Section 2, the field must be constant (following Poisson). Figure 3b is but an extension to sea level of an observation made by Le Mouël [20].

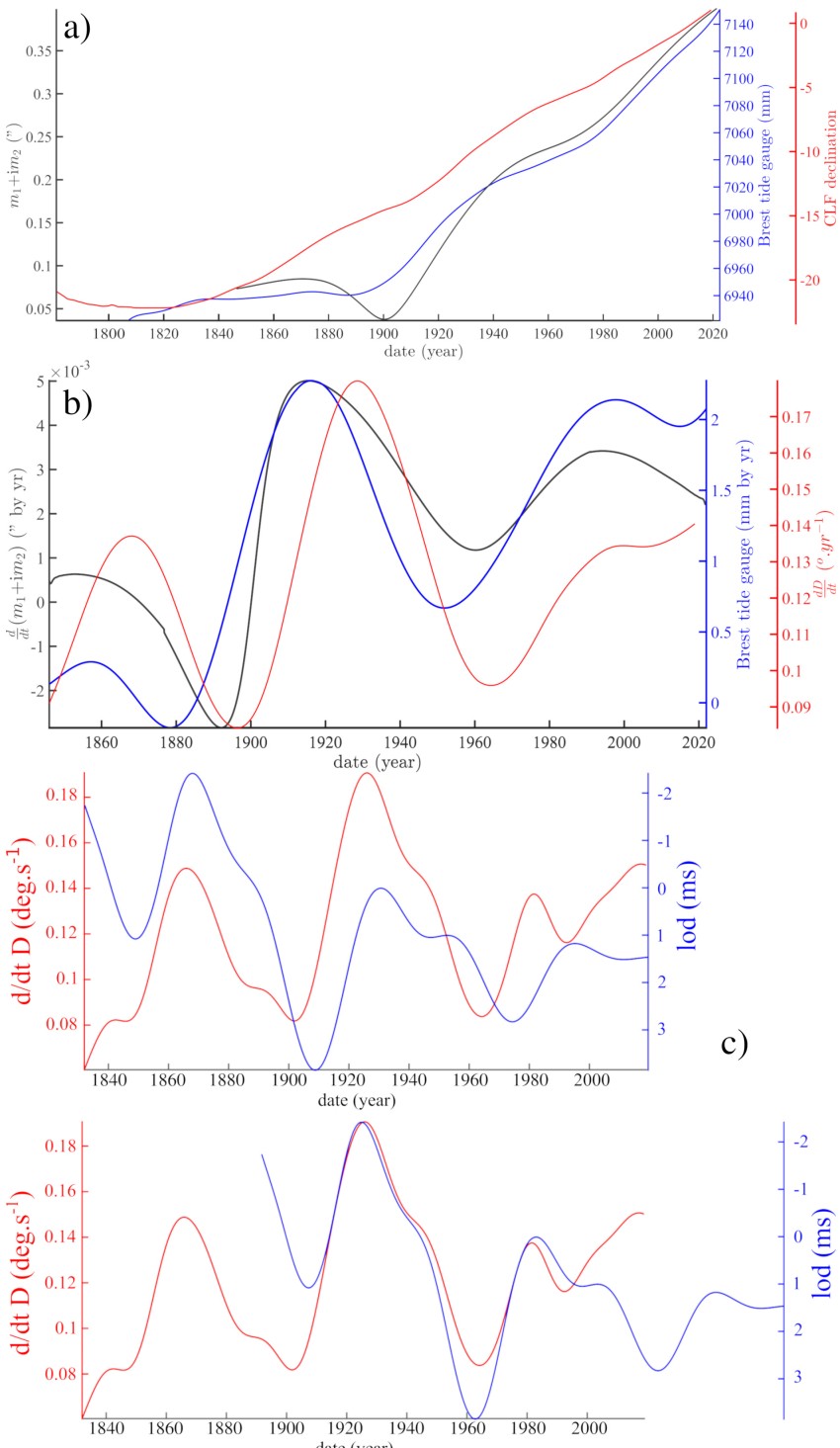

**Figure 3.** Various comparisons between the trends of magnetic declination in Paris, mean sea level in Brest, and polar motion and *lod*. (**a**) Superposition of SSA trends of (1) the Markowitz–Stoyko drift since 1846 (gray curve) (2) of the magnetic declination *D* in Paris since 1781 (red curve) and (3) of the mean sea level from Brest tide gauge since 1807 (blue curve). (**b**) Superposition of the first time derivatives of the three trends in Figure 3a. (**c**) On the top: superposition of the smoothed first time derivative of magnetic declination *D* in Paris since 1835 (red curve) and the smoothed length of day (blue curve) from Stephenson and Morisson ([68]) since 1835. On the bottom: the latter curve is offset by 60 yr.

### 4.2. On the Forced Quasi-Cycles of the Magnetic Field

Next, we extend the observation by Jault and Le Mouël ([24]) of a link between annual and semi-annual oscillations of lod and the magnetic field. We applied **SSA** in order to extract the annual and semi-annual components from the same three time series (sea level at Brest, magnetic field at CLF and length of day; this sub-section and Figure 4a–c).

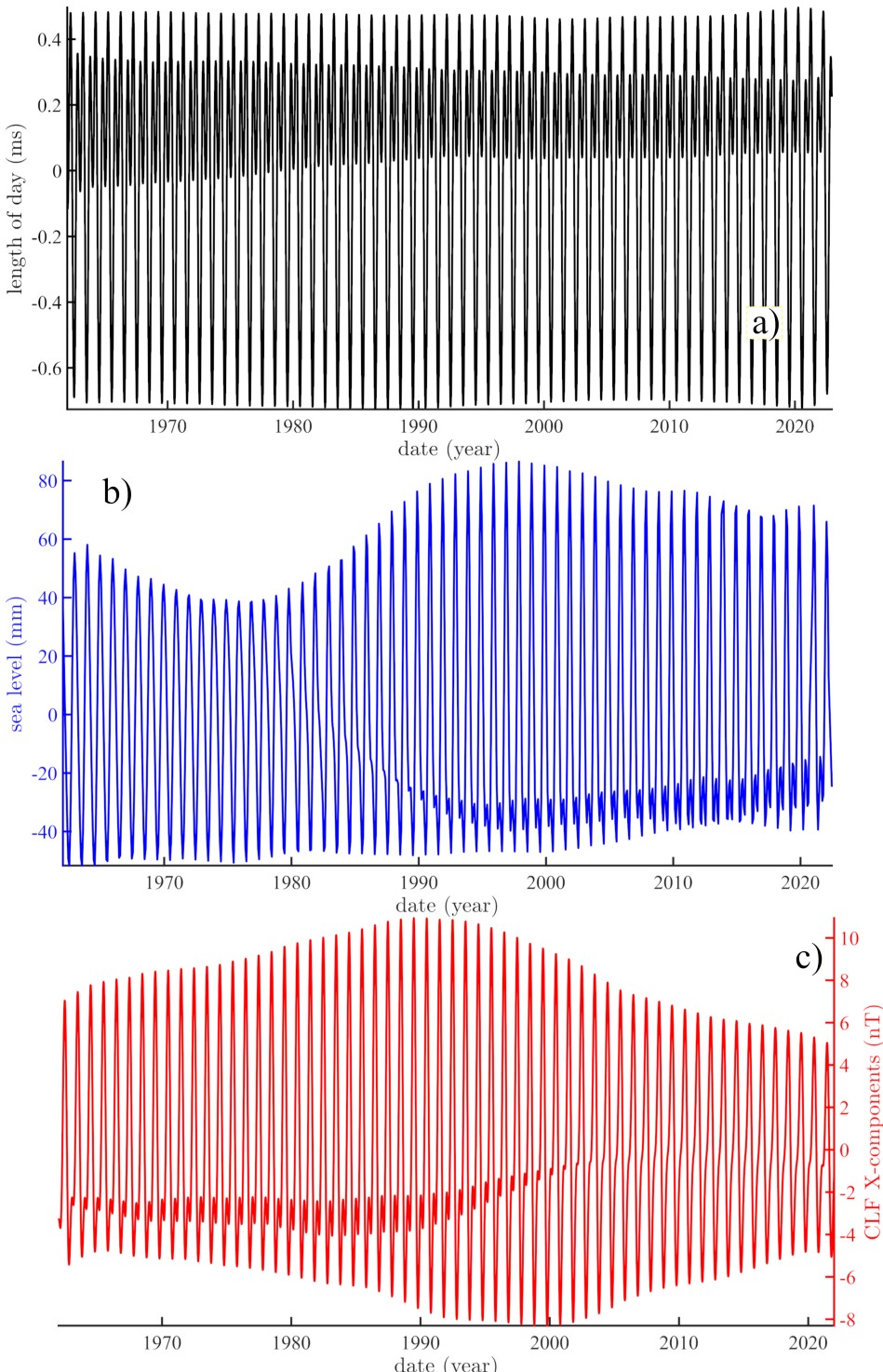

**Figure 4.** Sums of the annual and semi-annual components of (**a**) *lod* (**b**) Brest tide gauge and (**c**) the X magnetic field at CLF since 1962.

Given their phase and amplitude modulation, the superimposition of the semi-annual and annual components generates fringe patterns (Figure 4a–c). These are better visualized in the enlarged Figure 5a,b, showing the 1980–1990 decade. In Figure 5a, the two components generate a characteristic pattern with two humps that correlate between the magnetic field and length of day, with constant phases. In Figure 5b, one of the two humps for sea-level is subdued and looks more like a shifting step. The double hump pattern was already recognized by [24] as Figure 1 in their work. As explained by Poisson ([27]), a moving charged fluid tends to replicate the pattern of the motion; the different processes seem to be in phase and constant. If these originate from fluid motion tangential to the core, there will be a transfer of moment according to the Liouville–Euler equations. Thus, based on Figures 4 and 5, one can propose that the annual forcing of the double humps is driven by polar motion/rotation.

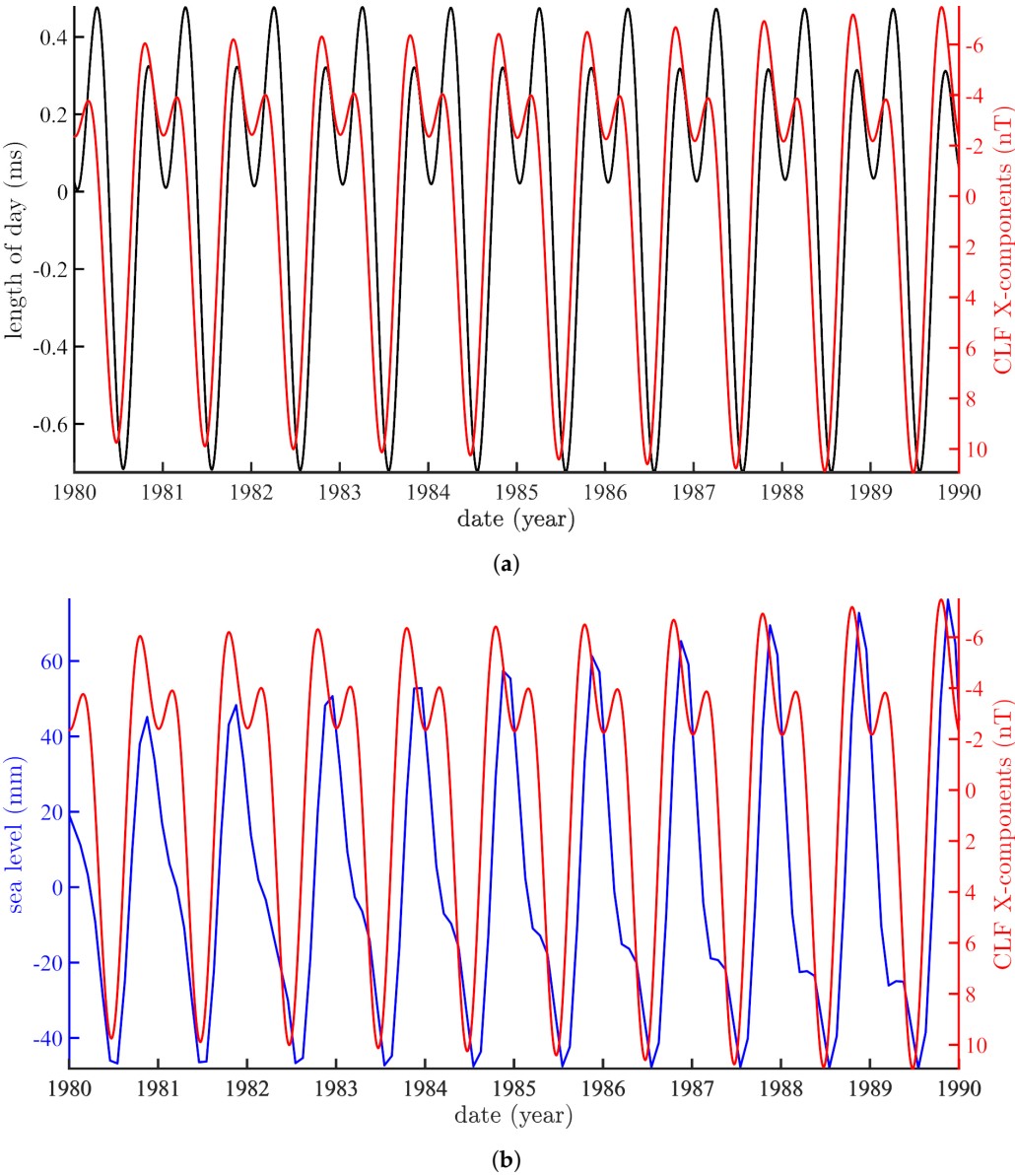

**(a)**

**(b)**

**Figure 5.** Enlargement and superposition of the annual and semi-annual components of the X magnetic component at CLF (red), *lod* (gray curve) and sea level at the Brest tide gauge (blue curve). (**a**) Length of day versus X component of the magnetic field in Chambon-La-Forêt (**b**) Sea level recorded by the Brest tide gauge versus X component of the magnetic field in Chambon-La-Forêt.

We next wish to check whether the same can be said of the other geomagnetic field components at CLF. Figure 6a,b compare the pattern of *lod* to those of X, Y and Z: Y behaves as X, but Z does not have the two humps, only one strong maximum per year (i.e., no semi-annual component).

We attempted to check whether the observations made with the couple "magnetic observatory–tide gauge" at CLF and nearby Brest could be extended to other couples. Unfortunately there are not many such couples, particularly since all datasets should be of sufficient length and quality. We found five couples listed in Table 1 below and shown on a world map (Figure 7).

**Table 1.** List of "magnetic observatory–tide gauge" couples.

| Magnetic Observatory | Tide Gauge |
| --- | --- |
| Chambon-La-Forêt (CLF, 2.26° E, 48.02° N) | Brest (4.49° W, 48.38° N) |
| Hartland (HAD, 4.48° W, 51° N) | Newlyn (5.54° W, 50.10° N) |
| Canberra (CNB, 149.36° E, 35.32° S) | Newcaslte V (151.78° E, 32.92° S) |
| Hermanus (HER, 19.23° W, 34.43° S) | Simons Bay (18.44° E, 34.18° S) |
| Kanozan (KNZ, 139.95° E, 35.25° N) | Mera (139.82° E, 34.91° N) |

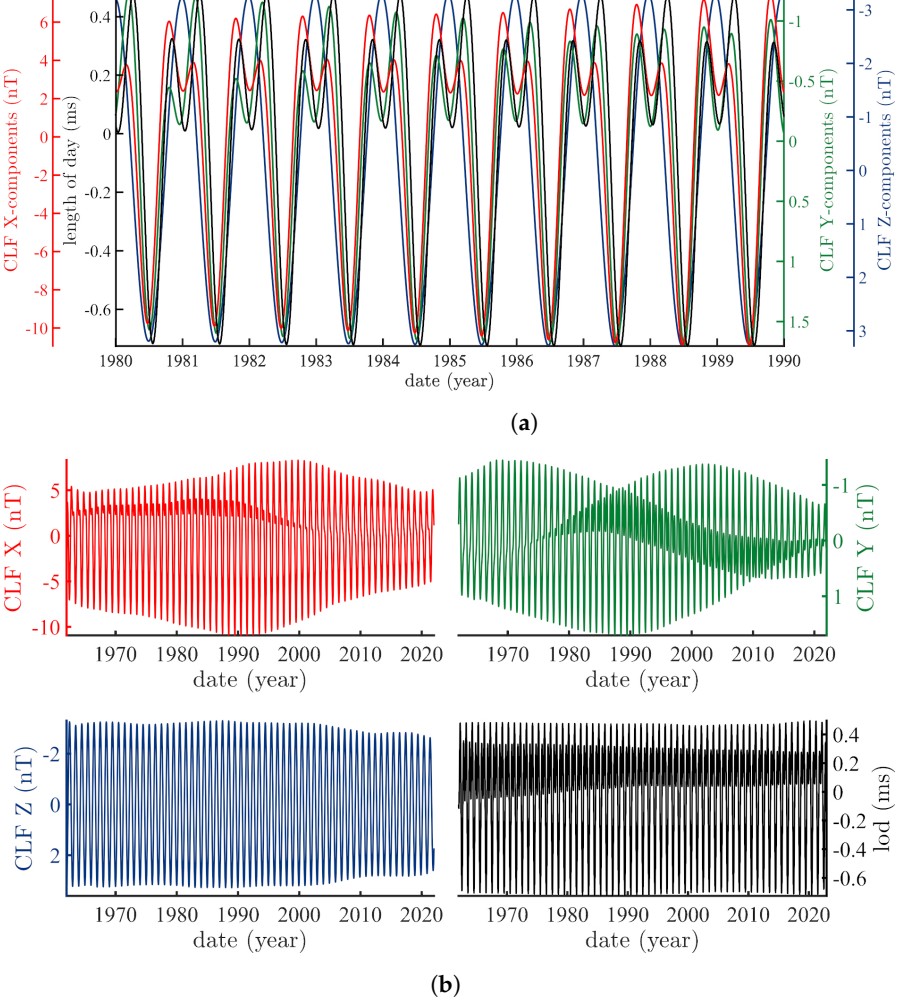

**Figure 6.** Comparison of annual plus semi-annual components of all three geomagnetic components at CLF with those of *lod*. (**a**) Comparison of annual plus semi-annual components of all three geomagnetic components X, Y and Z at CLF (X red, Y green, Z blue) with those of *lod* (gray) 1980–1990. (**b**) Annual plus semi-annual components of geomagnetic components X, Y and Z at CLF (X red, Y green, Z blue) and those of *lod* (1962–2022).

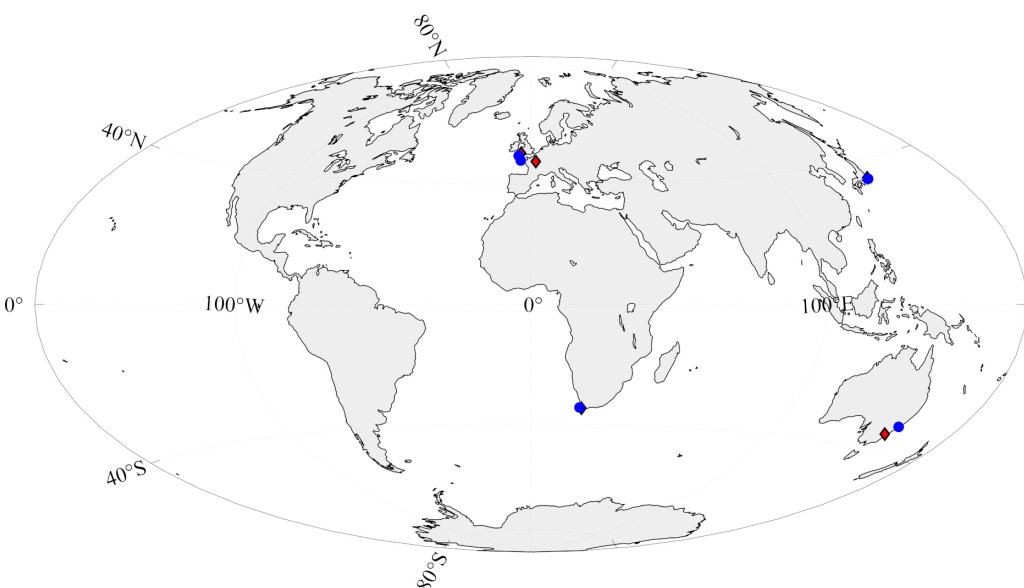

**Figure 7.** Associated couples of a magnetic observatory (red diamond) and a tide gauge (blue circles). See Table 1.

Figure 8a–c show the combined annual and semi-annual **SSA** components of X, Y and Z at the five magnetic observatories. These figures illustrate the different modulation ("wave") patterns associated with annual and semi-annual forcings. There is an ongoing debate as to their origins (e.g., [70–74]).

Figure 9a–e allow one to compare the magnetic oscillations for X, Y and Z with the corresponding sea level oscillation, one for each observatory couple. Magnetic components X extracted from the Brest-**CLF** (Figure 9a), Simons Bay-HER (Figure 9b) and Newlyn-HAD (Figure 9c) couples are in phase opposition with the sea level; the two other field components Y and Z are in phase with the sea level (with a small phase drift over the 40 years of the record). These three couples happen to be located on the same magnetic meridian. The same holds for the Newcastle-**CNB** couple (Figure 9e), with a slightly larger phase drift.

For the Japanese couple Mera/KNZ (Figure 9d), X is in phase with sea-level in 1980, when Y and Z are in quadrature. After 40 years of slow drift, Z is in phase, and X and Y in quadrature. Finally, for the Australian couple Newcastle/**CNB** (Figure 9e), X is in phase opposition, Y in phase and Z in quadrature in 1980 and the three drift respectively to opposition, quadrature and opposition in 2020. We note that in Hartland, Z does not have a semi-annual component (Figure 9c), and it is quite small in Hermanus (Figure 9b). We tested tens of potential tide gauge/magnetic observatory couples, whose results are not good enough to be reported; we just note that some 80% of them have no semi-annual Z.

The comparisons made in Figure 9 suggest strongly to us (though, granted, they do not prove) a link between annual variations of sea level and the magnetic field. The correlations are of better quality than we might expect, despite differences in topography and geography in the vicinity of the gauges. The same can be said of the (**SSA** determined) trends. We are in a position to strengthen our physical understanding of this link.

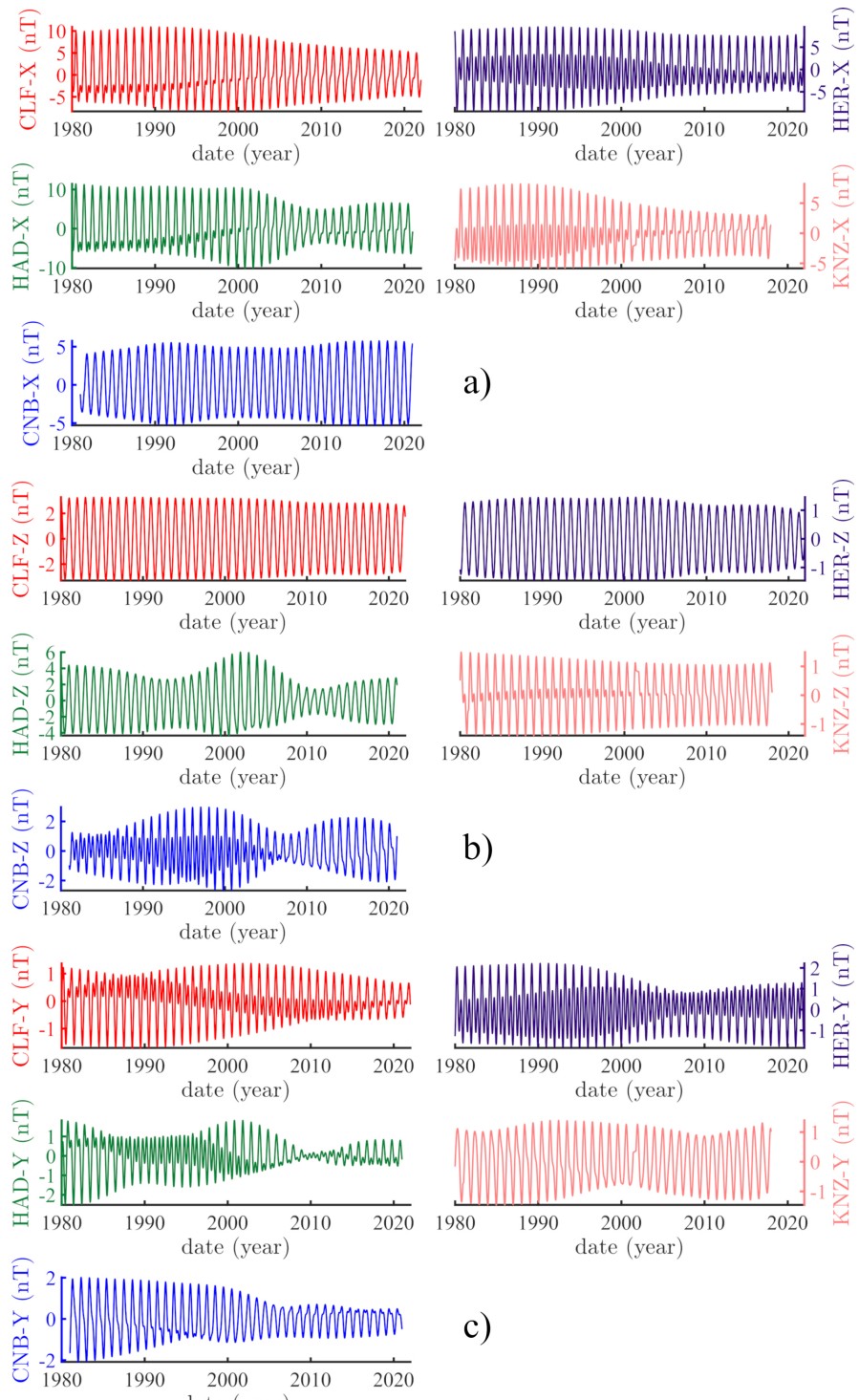

**Figure 8.** Time evolution of annual plus semi-annual components extracted from the 5 observatories listed in Table 1. (**a**) Forced quasi-cycles associated with the X magnetic component at (from left to right and top to bottom) Chambon-La-Forêt ( **CLF**), Hermanus (HER), Hartland (HAD), Kanozan (KNZ) and Canberra (**CNB**). See Table 1 and Figure 7. (**b**) Forced quasi-cycles associated with the Y magnetic component at (from left to right and top to bottom) Chambon-La-Forêt ( **CLF**), Hermanus (HER), Hartland (HAD), Kanozan (KNZ) and Canberra (**CNB**). See Table 1 and Figure 7. (**c**) Forced quasi-cycles associated with the Z magnetic component at (from left to right and top to bottom) Chambon-La-Forêt (**CLF**), Hermanus (HER), Hartland (HAD), Kanozan (KNZ) and Canberra (**CNB**). See Table 1 and Figure 7.

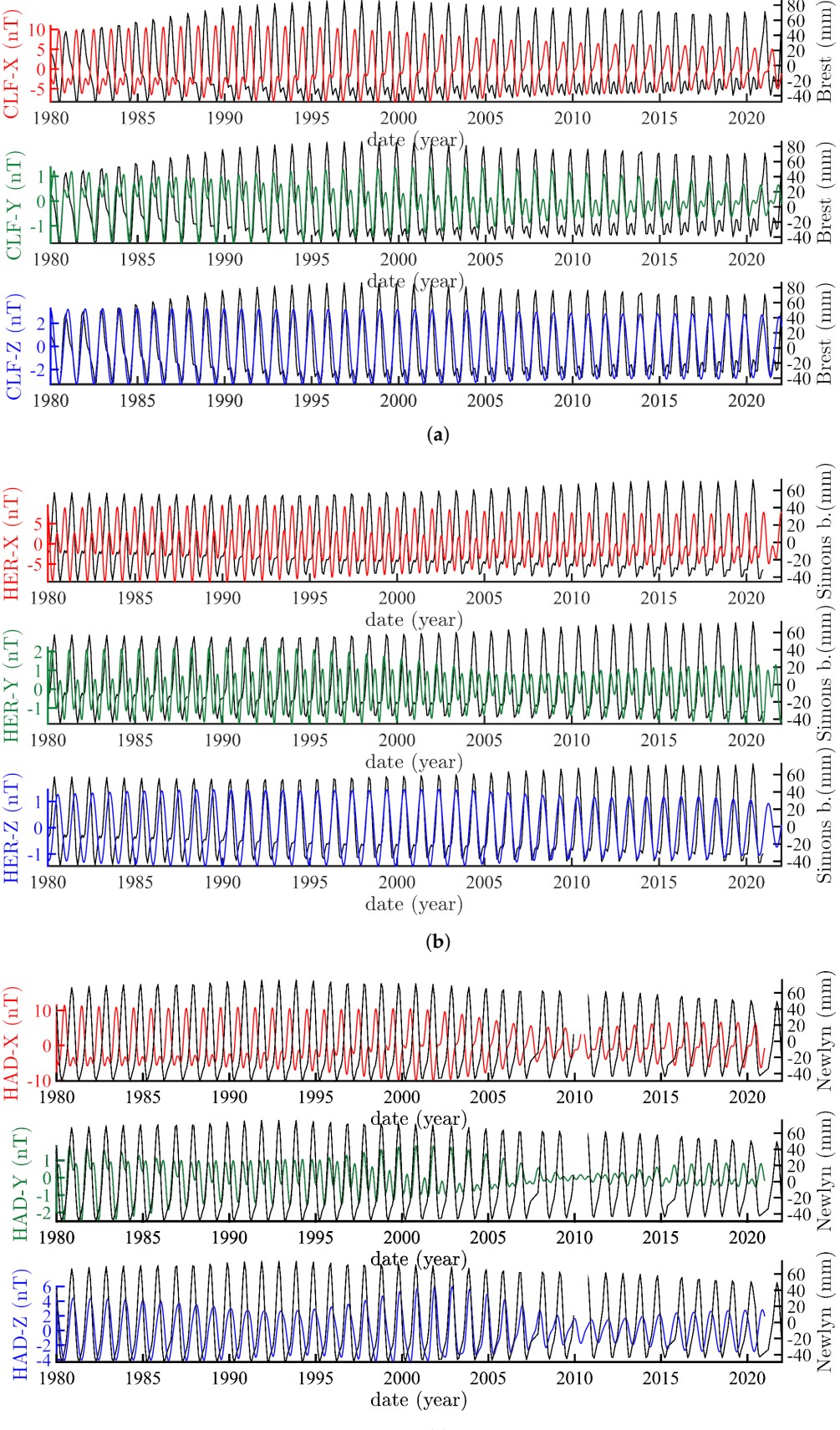

(**a**)

(**b**)

(**c**)

**Figure 9.** *Cont.*

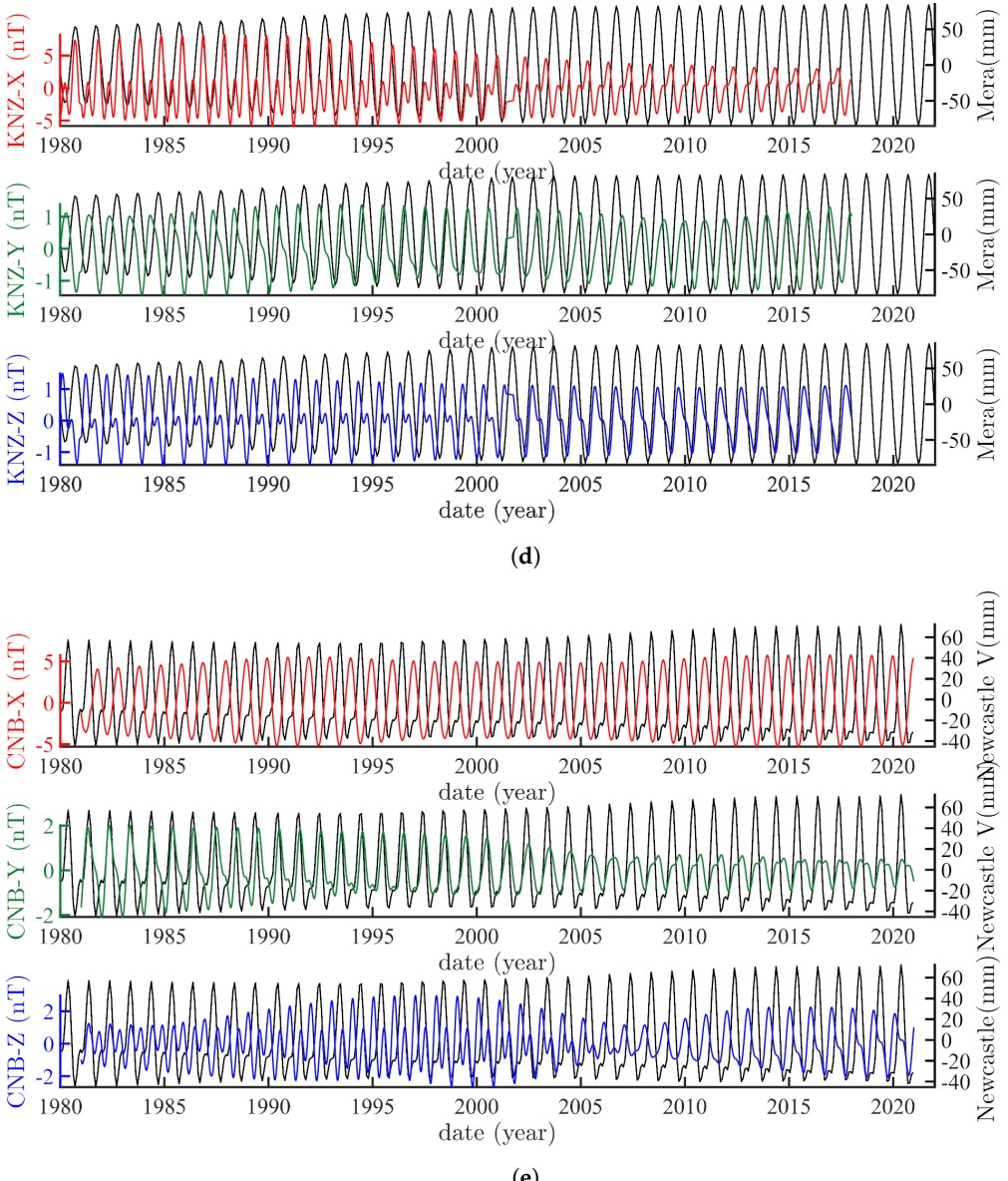

**(d)**

**(e)**

**Figure 9.** Comparison between forced components extracted from each observatory/tide gauge couple. (**a**) Comparison between forced components of sea level in Brest (grey curve) and respective oscillations of magnetic components X (red curve), Y (green curves) and Z (blue curve) in Chambon-La-Forêt. (**b**) Same as Figure 9a for the tide gauge/magnetic observatory couple Simons Bay/Hermanus. (**c**) Same as Figure 9a for the tide gauge/magnetic observatory couple Newlyn/Hartland. (**d**) Same as Figure 9a for the tide gauge/magnetic observatory couple Mera/Kanozan. (**e**) Same as Figure 9a for the tide gauge/magnetic observatory couple Newcastle/Canberra.

In Section 2, we saw that if a measured magnetic field has its origin in the uniform motion of charges in rotation about our planet's symmetry axis, the quantity and quality of charges cannot vary with time (or the dipole would vanish). Then, there is a link between the magnetic and angular moments. It is expressed by Equation (8a):

$$\mathbf{m} = \frac{e}{2mc} \sum \mathbf{r} \times \mathbf{p} = \frac{e}{2mc} \mathcal{M}$$

Sea-level and magnetic variations are linked through polar motion (e.g., [20], Section 2); here, the length of day. Polar motion is forced by Earth's revolution about the Sun. If the field is constant and the link exists, the ratio $\frac{e}{2mc}$ must be constant. The (geo-)physical

consequences generated by these moments should be, of the first order, the same for all torques around the globe, and the variations in amplitude of the geophysical phenomena involved should be proportional.

In Table 2 below, we evaluate the amplitudes of the **SSA** annual components of the sea-level and magnetic components and their ratios for the five couples of stations of Figure 7. In the 1980–2022 period, these ratios are the same at 7 mm/nT for CLF, HAD and HER, almost the same at 8 mm/nT forCNB and not so different at 10 mm/nT for KNZ. Given the complexity of sea-level physics and geomagnetism, there is no a priori reason why the ratios should be constant, unless Equation (8a) holds, which seems to be the case. This result vindicates Poisson's approach ([27]): fluid motions in the core are similar to those at the surface; because they are charged, the motifs of variations in the Earth's magnetic field are those of the sea surface and the atmosphere.

**Table 2.** List of "magnetic observatory–tide gauge" couples (a completer).

| Couple Observatory–Tide Gauge | Ratio Sea Level/Magnetic Component | Order of Magnitude |
| --- | --- | --- |
| CLF/Brest, between 1980 and 2000 | ∼100 mm/15 nT | ∼7 mm/nT |
| HAD/Newlyn, between 1980 and 2005 | ∼100 mm/15 nT | ∼7 mm/nT |
| CNB/Newcaste V, between 1980 and 2022 | ∼ 80 mm/10 nT | ∼8 mm/nT |
| HER/Simons bay, between 1980 and 2000 | ∼100 mm/14 nT | ∼7 mm/nT |
| KNZ/Mera, between 1980 and 1995 | ∼140 mm/14 nT | ∼10 mm/nT |

*4.3. On the 11 yr Cycle and the Magnetic Field*

The 11 yr cycle is one of the better known variations of the Earth's magnetic field. It is considered to be the same Schwabe ([75]) cycle that is found in sunspot numbers. In the minds and logic of Laplace ([33]), Lagrange ([36]) and Poisson ([27]), planets are responsible, through exchanges in moment, for a number of astrophysical and geophysical phenomena. The origin of this 11 yr cycle likely has its source in the revolution and moment of Jupiter. This moment is directly connected to variations of distances to Jupiter in the solar system. Jupiter exerts the largest torque of the eight planets (and also larger than that of the Sun, which is motionless at the time scales we are interested in; Lopes et al. [67]). This torque acts on (or modulates, or forces) sunspots (e.g., [55]) as well as on polar rotation (e.g., [49,66]).

In what we will call the Poisson–Le Mouël paradigm, the fluid's rotation generates the field; this same rotation must then force solar activity in the form of sunspots. In the case of fluid mechanics, and as shown by Courtillot and Le Mouël ([76]) Figure 45 and by Le Mouël et al. ([6,18]), a law of natural turbulence appears, a Kolmogorov ([19]) power law with exponent $-5/3$. This law is found in sunspots as well as in variations of geomagnetic intensity (the latter since as far back as at least 1 Myr, see [76]). The angular coordinate $\theta$ and $\frac{d\psi}{dt}$ that define the location of the pole of rotation on the sphere are the solutions of a system of differential equations, one for $\theta$ that describes pole motion, and the other for $\frac{d\psi}{dt}$ that describes the length of day. One is the derivative of the other (see [51]). As is the case for all derivative operators, it amplifies high-frequency components. Le Mouël et al. ([77]) showed that the 11 yr component is a major component of the length of day, whereas Lopes et al. ([66]) showed that it is a minor one of polar motion.

In order to retain the homogeneity of our datasets and their temporal resolution, we integrated the 11 yr quasi-cycle extracted from the length of day by **SSA** (Figure 10, black curve, bottom two rows). In the top row of Figure 10, we superimposed the 11 yr component from sunspots (pink) with that of *aa*. The two curves appear to be in quadrature: this is checked by offsetting the Schwabe cycle forward by exactly the 11/4 yr (second row). The explanation for this observation is the following: the torque exerted by Jupiter acts directly on sunspots, while the aa index is the difference between two antipodal observatories. Thus, the *aa* index is a derivative operator. This is likely why the 11 yr cycle is prominent in *aa* but minor in the X, Y and Z components. The same accounts for the phases of aa and

lod: we integrate the 11 yr component of lod (black curve in 3rd row of Figure 10) and see it is in phase with aa. And finally, according to [36], Jupiter does act on the Earth's rotation, as shown by Lopes et al. ([49]) and the Jupiter–Earth distance (blue curve bottom row in Figure 10). This last result provides a good illustration of Equation (8f):

$$\frac{d\mathcal{M}}{dt} = -\Omega \times \mathcal{M}.$$

Polar motion as well as length of day are linked to $\Omega$ (e.g., Lambeck [56], Section 3). We have also seen in theory (and checked in the example above) that the variation in the magnetic field is linked to moment $\mathcal{M}$. We could say that magnetic field components (X,Y,Z) are to polar motion ($m_1$,$m_2$) what *aa* indices are to the length of day. The link between Sections 2 and 3 above is the length of day.

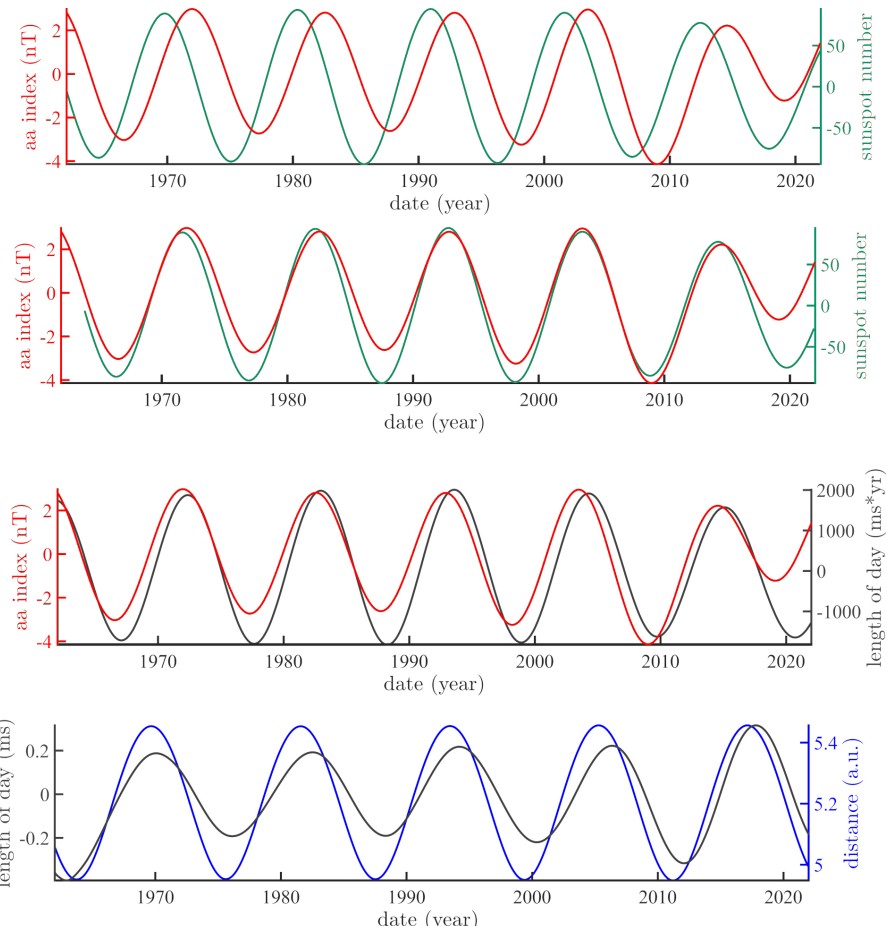

**Figure 10.** Eleven year quasi-cycles extracted by SSA from the geomagnetic index aa (red; top 3 rows), the sunspot series (pink; top 2 rows), the length of day (black; bottom 2 rows) and the ephemerids of Jupiter (blue; bottom row marked as distance of Earth from Jupiter).

### 4.4. On the International Geomagnetic Reference Field

Since the work of Gauss ([28]), the decomposition of the geomagnetic field into spherical harmonics has become normal practice ("routine"). We recall, however, as a caveat the fact that an electric field or a gravity field can be decomposed in spherical harmonics (SH) because these fields are constant and their elementary sources all have the same sign. Such is not the case with a magnetic field, unless it is constant also (Sections 2 and 3). In principle, the SH decomposition of a magnetic field does not have physical significance. Nevertheless, a spherical harmonic decomposition of the International Geomagnetic Reference Field (**IGRF**) is published every five years ([78]). Given the fact that there is no magnetic monopole, the first source term (i.e., "Gauss coefficient") is the axial dipole $g_{1,0}$,

an imaginary source at the center of the Earth. The other terms of the Fourier expansion on the base functions cos and sin are written as $g_{l,m}$ and $h_{l,m}$.

Figure 11a shows the monotonous decay of the **IGRF** axial dipole $g_{1,0}$ since 1900. With Poisson ([27]) and Le Mouël's ([20]) hypothesis in mind, and given some of the results in the previous sections of this paper (similar behavior of the annual **SSA** components of sea level, rotation axis and magnetic field), it is natural to compare the behavior of the intensity of the **IGRF** dipole with polar motion $(m_1, m_2)$ or the equivalent parameter $\theta$ of [33]. This is done in Figure 11a.

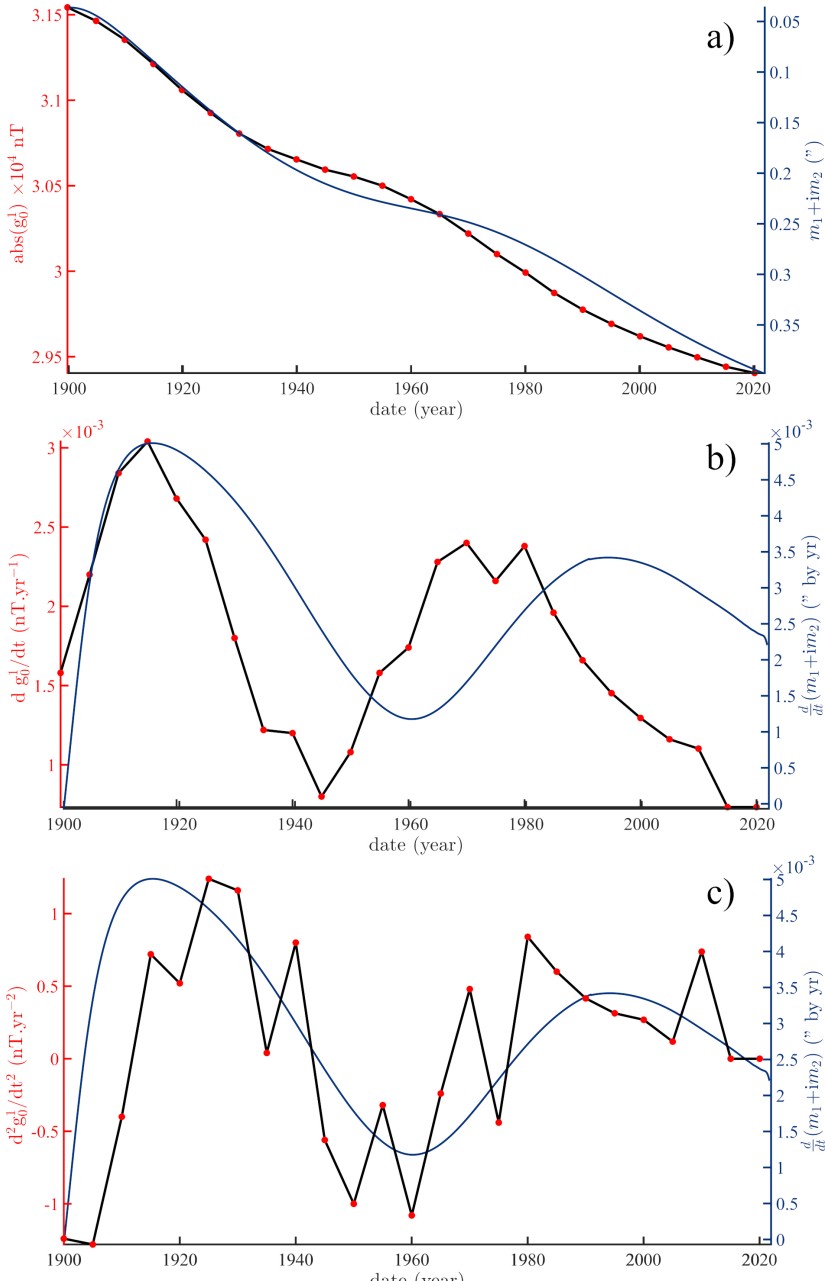

**Figure 11.** Comparison between (**a**) the evolution of $g_{1,0}$ from ([78]) and the Markowitz–Stoyko drift and (**b**,**c**) their corresponding time derivatives. (**a**) Red dots = **IGRF** $g_1^0$ every 5 years since 1900 [78] and interpolation as the black curve. Blue curve = **SSA** first component i.e., trend of polar motion, that is the Markowitz–Stoyko drift. (**b**) First time derivatives of the two curves shown in Figure 11a. (**c**) Second derivative in time of the curve corresponding to the coefficient $g_1^0$ versus the first derivative of the polar motion. The two curves are in phase possibly due to causality.

The time variations of $g_1^0$ and $\theta$ are indeed very close. The first derivative of polar motion is in quadrature with the first derivative of $g_1^0$ (Figure 11b) and in phase (opposition) with the second derivative of $g_1^0$ (Figure 11c).The variation of dipole intensity follows the variation of the angle $\theta$ between the rotation axis of the planet and that of the dipole. Another example is given by the similarities between the magnetic declination in Paris and rotation axis, and between their derivatives (Figure 3a,b).

If the field is constant, we saw that there is a link between the magnetic moment **m** and the angular momentum $\mathcal{M}$ (Equation (8f)). We checked this relation, using observations (Table 2). We also showed that, in the case of a rotating system of charges about the axis $Z$ (with velocity $\boldsymbol{\Omega}$), this relation becomes

$$\boldsymbol{\Omega} = \frac{e}{2mc}\mathbf{H}$$

This does express the link between the variations of Earth's rotation and the magnetic field **H**. Since $\boldsymbol{\Omega}$ is connected to $m_1$ and $m_2$ through the Liouville–Euler equations, **H** is also connected to $m_1$ and $m_2$. But the key coordinate is $\dot{m}_3$, which is the length of day. This is the reason why the second derivation of $g_{1,0}$ agrees better with the Markowitz–Stoyko drift, which is, as we saw, the second derivative of *lod*.

## 5. Discussion and Conclusions

In his pioneering work on the nature and origin of the Earth's magnetic field, Poisson [27] recognized that the field had to be constant. Gauss [28] came to the same conclusion. This will seem awkward to the modern physicist. Poisson's proof involved theoretical physics and mathematics but only a limited set of observations.

More recently, one of us (Le Mouël [20]) observed that there were strong observational connections between the Earth's magnetic field and its rotation, more precisely, the secular variation of the field and the drift of the rotation pole on one hand, and the forced oscillations of the magnetic field and the length of day on the other. Based on these observational facts, Le Mouël [20] proposed to model the core source as a rotating cylinder when Poisson envisioned a sphere. Thus, based on observations, Le Mouël (1984) (with collaborators, in a suite of often cited papers) built quasi "experimentally", 150 years later, almost the same theory as Poisson had done "theoretically".

We attempted in this paper to return to the original sources and to reconstruct the development of geomagnetism, using the "current" language. Starting from Maxwell's equations, we derived the equations for the electrostatic and magnetostatic fields (Section 2). We came to the same equations as Poisson, with an important difference: Poisson chose the Lagrangian approach to gravity (Legendre, 1785; Lagrange 1788; Laplace, 1799) to formally derive the equations for the magnetic field ([35]). To scientists of this epoch, there was no difficulty in reasoning the magnetism in the same "classical" way as one reasons in gravity.

In Section 2, we propose a short tutorial on the equations of electromagnetism. This can be found in most graduate textbooks. But we emphasize the fact that the magnetic field does not need to derive from a scalar potential to be developed into spherical harmonics. However, most geomagnetists do make this hypothesis, invoking Stokes's theorem: since magnetic measurements are made at the surface where almost no charge circulates, one can assume that the field derives from a scalar, not a vector potential. It is more physical, hence logical, to obtain the spherical harmonics from the electrostatic field, then use the vector potential (Equation (6)) to return to their expression for the magnetic field.

One can perform a decomposition in spherical harmonics and consider multi-poles if and only if the motions of charges are finite and uniform, two conditions that are not met in a dynamic field. From this, one can draw a number of consequences for the magnetic field, the main one being that the magnetic moment of the charges that generate the field and their angular moment (thus, the motion of the rotation pole) are linked by Larmor's relation. This is in agreement with the theoretical works of Laplace [33], Poisson [27]) and [20]. A magnetic field

can be written on the basis of spherical harmonics only if this field is constant. Section 3 deals with the intrinsic theoretical consequences of a constant dipolar field.

From the ideas presented in Sections 2 and 3, we concluded that in order to satisfy all hypotheses and all theoretical results, it is sufficient that the axis of symmetry of the magnetic dipole and the rotation axis can move with respect to one another, which is never the case with a development in spherical harmonics.

Section 4 presents tests of our hypotheses. In Figure 3, we show that sea-level variations in Brest, variations of magnetic declination in Chambon-La-Forêt and Markowitz–Stoyko polar drift were essentially the same, except for the phase of declination, which we interpret as resulting from the nature of core–mantle coupling. We know from other tide gauges that the Brest gauge is representative of most northern hemisphere gauges. Jault and Le Mouël (1991) discussed the correlation and the model that would link geomagnetic secular variations and polar drift; we extend the correlation and the model to sea-level variations.

Next, we focused on annual and semi-annual oscillations, as did Jault and Le Mouël [24]) previously, bringing into the picture the information carried by the sea level. All these cycles, regardless of whether they correspond to the geomagnetic field components, length of day, or sea level, are in phase or in phase opposition. This observation falsifies the hypothesis that the field would include seasonal variations (the Russel–McPherron [72] effect), unless the effect could also explain the presence of the same cycles in the sea-level and length-of-day variations. We calculated the ratios of the mean amplitude of variations in these cycles (magnetic field components to associated tide gauges). Independent of local and regional geography and topography, we find these ratios to be quasi constant at ∼8 mm/nT. This also agrees with the concept of a constant field (relation (8a)).

We next tested the Schwabe ∼11 yr cycle. Paradoxically, it is rather weak in geomagnetic field components, and quite strong in magnetic indices, such as aa. In a parallel way, the cycle is weak in polar motion, yet it is one of the main components of the length of day. This is readily understood when one follows Laplace (1799): polar motion and length of day describe the same phenomenon but differ by one order of derivation. In a reciprocal way, the polar motion is linked to the derivative of magnetic components.

Our last test is with the secular variation of the IGRF. Many forget the caveat that a dynamic magnetic field derives from a vector potential, not a scalar potential; hence, it can, in principle, not be developed in spherical harmonics. Despite this caveat, regular analyses of magnetic field models have been produced at five-year intervals for years since 1900. The secular variation of that field is monotonous and decreasing as far as the leading axial dipole field component is concerned. This behavior is parallel to that of polar motion. We saw, in addition, that the time variations of the second derivative of $g_1^0$ are parallel to those of the first derivative of polar motion.

The validity (and foresight) of Poisson's derivation of Maxwell's equations also applies to Larmor's equation, the Liouville–Euler system, and, in general, the distinction that must be made between electric and magnetic fields. The parallel behaviors of magnetism and rotational mechanics, illustrated by the Larmor formula, were put to the test with modern observations with success. We successively showed that the drift of the magnetic dipole, the forced responses of the field to commensurate pseudo-frequencies generated by the Jovian planets, the Schwabe cycle and the IGRF, can all be understood in the frame of Poisson's theory. And to the series of observations of a number of geophysical, atmospheric and heliophysical processes (e.g., [53–55]), including sea level, the Earth's magnetic field can be added as we showed in this paper.

**Author Contributions:** V.C., J.-L.L.M., F.L., D.G. and J.-B.B. contributed to conceptualization, formal analysis, interpretation and writing. All authors have read and agreed to the published version of the manuscript.

**Funding:** This research was supported by the Université de Paris, IPGP, the LGL-TPE de Lyon and Museum National d'Histoire Naturelle.

**Data Availability Statement:** The used data are freely available at the following address: IERS: https://www.iers.org/IERS/EN/DataProducts/EarthOrientationData/eop.html (accessed on 12 March 2023); Permanent Service of Mean Sea Level: https://www.psmsl.org/data/obtaining/complete.php accessed on 12 March 2023; Geomagnetism Data Portal: https://wdc.bgs.ac.uk/dataportal/ accessed on 12 March 2023.

**Acknowledgments:** We thank the three anonymous reviewers for their careful critical reading of the paper that led to many corrections and improvements and, we hope, the better readability of the revised version of the paper.

**Conflicts of Interest:** The authors declare that they have no known competing financial interest or personal relationships that could have appeared to influence the work reported in this paper.

## Appendix A. The Polar Motion

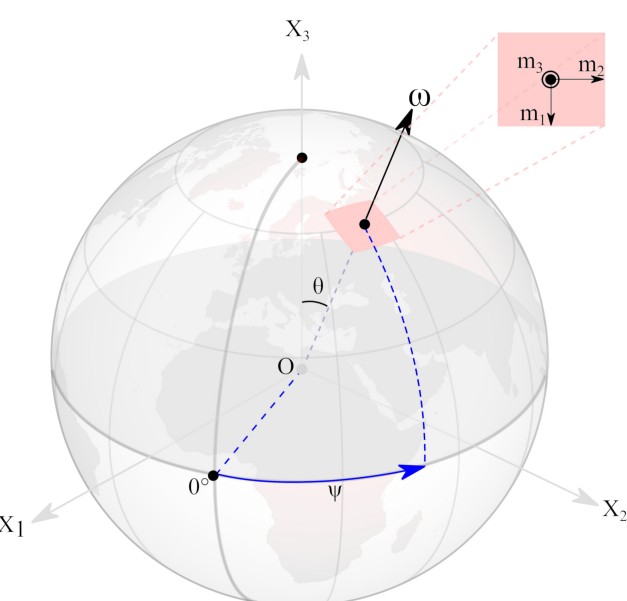

**Figure A1.** Terrestrial reference frame. $m_1$ and $m_2$ are the coordinates of the rotation pole. $\psi$ and $\theta$ are the declination and inclination introduced by Laplace [33]. From Lopes et al. [67].

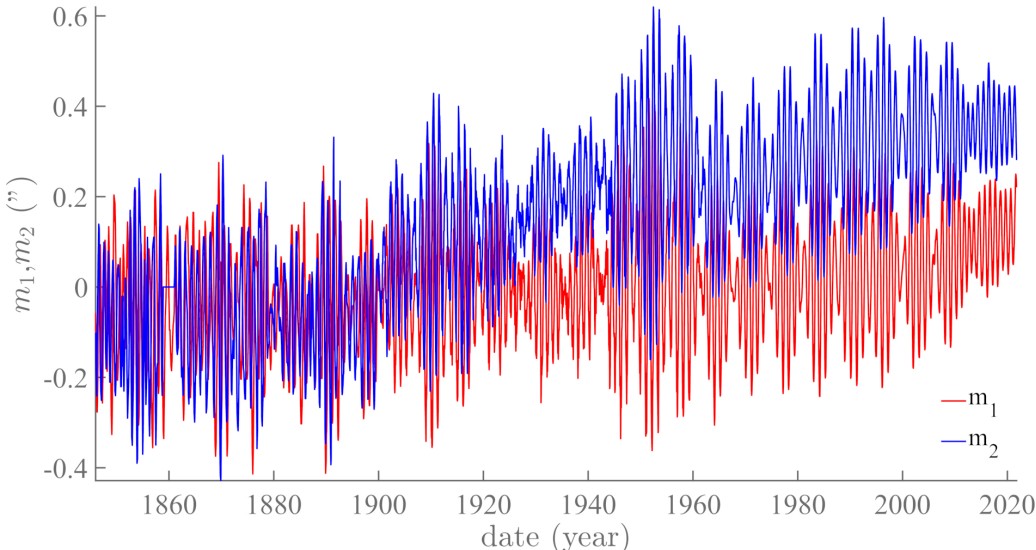

**Figure A2.** Polar motion from IERS. The $m_1$ and $m_2$ components of polar motion from IERS (from 1846 to the present). From Lopes et al. [67].

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
