# Peer review of "Is the Earth’s Magnetic Field a Constant? A Legacy of Poisson"

_geosciences, doi:10.3390/geosciences13070202_

Round 1

Reviewer 1 Report

This manuscript of 37 pages is rather attractive if one looks at the title and the abstract which is clear. However, the mathematical development of a theory which can be found in many undergraduate textbooks (according to the authors) is rather difficult to follow partly because of the  used notations and symbols. At many parts of the text, bold symbols (to represent vectors) for scalar quantities and normal symbols for vectors.  The number of figures and their quality are not a good standard. The feeling is that the reader is lost in this long manuscrip. Some figures are plotted on more than one page, The authors should focus on the link between the physical quantities indicated in the abstract, reduce the length of the manuscript and improve the quality of the figures, equations and manuscript. I don't understand why the authors refer to about 25 refs of their own work among 79 references.

Below some comments.

-- Page 2. Explain for what CMB stands for.

--Page 3. L82-86. A long sentence which is not clear.

* Why EM is in bold?

* Usually one used bold B to designate the magnetic field. Here the authors use bold H for the magnetic field.

* L 95. c is the velocity of the charged particle. Does not c stand for the speed of light?

* L109. which field is concerned (magnetic or electric or both)?

--Page 4.

* why don't you indicate that you use the CGS system?

*L126. What does df stand for?

* In eqs (2e) and (2f), R and R_i should not be bold. They should be scalar like the potential \phi.

* L127. (2f) represents a scalar potentiel not a field is indicated in this line.

* In eq (3a) what does e_a stand for?

--Page 5.

* In eq (3b) appears e_i while in eq (3a) e_a is used.

* L133, k is a vector.

* eq (3c) R_0 should be a scalar (gradient of a scalar).

** Why don't you use arrows to represent vectorial quantities?

* In eq (3d) how do you obtain the right expression from the left one?

* The line before eq (4a), R_0 and R_0^{n+1} should not be bold.

-- Page 6.

* Eq (4b) and othrers. Over what the summation is done?

--Page 7.

In eq (5d), the sum over m is from -l (letter L) not from value -1 (minus one).

* Eq (5e). Are you sure about capital \theta and \phi?

--Page 9.

* The magnetic moment m should be a vector (bold or with an arrow). Same remark in other parts of the manuscript.

* The development of rotational \vec{a} x \vec{b}, use bold or arrow to represent vectors.

* L164. Why don't you use \vec{L} to represent the kinetic moment. Please keep calling it kinetic moment instead of mechanical moment.

-- Page 10.

Third line. what does 5ft stand for?

-- Page 15. Indicate that "lod" is the abbreviation of Length of the Day.

-- Page 29. The pink and red colors are not too diffrent. Please use a different line style (dash, dot....)

-- Page 30. Please indicate that SH is the abbreviations of Spherical Harmonics.

Is it necessary to use all these references? 25 of your own work?

Author Response

Please find our answers in the uploaded pdf as well as in the new version of the manuscript.

Reviewer 2 Report

Review of "Is the Earth's magnetic field a constant? a legacy of Poisson" by Mouel et al.

This paper is proposed to test the surprising assertion for the first time evoked by Poisson (1826) using actual magnetic field data and IGRF model field. The authors have chosen to compare (1) the polar motion drift and the secular variation of the Earth's magnetic field, (2) the seasonal variation of day length together with those if the sea level recorded by different tide gauges around the globe and those of the Earth' magnetic field recorded in different magnetic observatories. This topic is interesting, but the readers can not easily understand what is significance, the future perspective of this study, and contribution to other research field . The authors should clarify these items before publication in this journal. The revision points are shown below.

1. Line 13: IGFR --> IGRF

   Please spell out this abbreviation.

2. Line 95:  Is c the velocity of the charged partilce?

     I think that this value is the velocity of light.

3. Equation (3a)

    ea --> ei?

4.  In Figures 3 to 10, several legends are very thin (especially, the gray characters). Please change the representation of these legends.

5. The section 5 (conclusion) is much longer, and the reader can not easily understand what is clarified in this study, what is significance, contribution to other research field. Please rewrite this section to easily understand the above items. I also suggest that the authors should add discussion section in this manuscript. In the full paper, this section is very important.

6. Please write the Acknowledgments of this paper. In the current version, only "to do" is described.  

Author Response

(The authors gave the same response as above.)

Reviewer 3 Report

In this manuscript (ms), the authors discuss Poisson’s suggestion of the origin of the Earth’s magnetic field. Since Poisson and Gauss both considered the magnetic field to be constant, the authors test this assertion with the most recent data. Firstly, they present a development of Maxwell’s equations in the framework of static electric and magnetic field in order to draw the necessary consequences for the Poisson hypothesis, and then check whether observations agree with this hypothesis. Secondly, the authors propose a mechanism, in the spirit of Poisson, to explain the presence of the 11-year cycle in the magnetic field. Finally, they study closely the evolution of the time evolution since 1900 of the g_{1,0}  coefficient of the International Geomagnetic Reference Field.

                The ms is extensive, well written, and presents very interesting Physics in the heart of classical electromagnetism. The conclusions are compatible with the results presented and in general this ms is of very high quality that certainly deserves publication in Geosciences. The presentation, however, can be improved in several points which are mentioned below:

1)      For the readers better information, in the first sentence of the second paragraph of the Introduction on page 2, ll.38-41, “In the three centuries … and its sudden jerks ([7–9])” the authors should also mention that the magnetic field of the Earth also exhibits variations before earthquakes ([Seiya Uyeda and Haruo Tanaka 2004 “Physics in Action: Maxwell's equations and earthquakes”,  Phys. World 17 (2) 21, https://doi.org/10.1088/2058-7058/17/2/30 ; P. A. Varotsos, N. V. Sarlis, and E. S. Skordas 2003 “Electric Fields that “Arrive” before the Time Derivative of the Magnetic Field prior to Major Earthquakes” Phys. Rev. Lett. 91, 148501, https://doi.org/10.1103/PhysRevLett.91.148501 ])

2)      In line (l.) 92, a reference to a textbook would be helpful for the reader, e.g., ([L.D. Landau, E.M. Lifshitz,  Course of Theoretical Physics, Volume 2, The Classical Theory of Fields (Fourth Edition), Pergamon, 1975, Pp. 402, ISBN 9780080250724, https://www.sciencedirect.com/book/9780080250724/the-classical-theory-of-fields ]).

3)      Since in modern textbooks Electromagnetism is taught in Systeme Internationale, it would be helpful to state after Eq.(1b) “written in CGS.”

4)      I cannot follow the reasoning expressed in the first paragraph of page 10. The authors should elaborate more. Why d/dt [Sum(e/c r x H)] is zero for an arbitrary distribution of charged particles in a constant magnetic field.  

5)      In the line after l.173, the term angular momentum is more common than “kinetic moment”

6)      The general audience of Geosciences would benefit from a more detailed definition of m_1 and m_2 in l. 251 probably through a figure.

7)      The style of figure captions 3, 6, 8, 9, and 11 is not compatible with that of the journal. Please include the captions of the panels to a single figure caption.

8)      In Figure 10 in the top two panels, the pink color is not discernible. Please use a different color.

9)      Acronyms should be defined the first time they appear both in the abstract and in the main text. Hence, IGFR should be defined in l. 13 of the abstract, CMB in l.53, SH in l. 388.

10)   Typos should be eliminated, in l. 12 “11-year cycle in”, l.64 “([28])”, Eq.(3d) the authors should check the subscripts, two lines before Eq.(4c) the inverted arc symbol before r squared should become minus sign “-“, third line of page 10, “5th equation”, l.193 “(5a)”, ll. 212, 213 the inverted arc symbol before r squared should become minus sign “-“, l.282 “8a)”, l. 283 “8f)”, p.20 in the second line of Figure caption 5 “curve) and b) sea”, l.354 “(8a)”, last but one line of page 29 “(8f):”, three lines after l.408 “(Table 2). We”, l.479 please check 8a, l.519 “Peragrinus, P.” l.520 “1917. ”.

Hence, I will be glad to suggest publication of an appropriately improved ms along the lines mentioned above.

Author Response

(The authors gave the same response as above.)

Round 2

Reviewer 1 Report

The authors did good job by amending their original manuscript. They have clarified all the points I have raised in my first report and I think also to those of the two other reviewers. I think this revised version of the manuscript is of a good quality even though I am still concerned by the unusual extension of several figures on more than one page. I recommend either to reduce the size of subfigures in such a manner that they hold in a single page with their caption as well. The other option consists in dividing these figures covering more than one page into two figures. However, this option has the disadvantage of increasing the number of figures.

There are still some corrections to be done prior to publication of the manuscript.

- Page 6, the added sentence "the letter t is missing in "the" sum being etended to all charge", also charges should be plurial.

- Page 7

** In eqs (5d) and (5f), the sum should be from -l (minus letter small L) to +l (small letter L). This error has already been pointed out in my first report and the authors said they have corrected but in reality not.

-Page 9 (and elsewhere).

* Angular momentum vs kinetic momentum. It is true that in my first report I suggested to the authors to use the term "kinetic moment" instead of mechanical moment". However, as suggested by reviewer #3, the more appropriate term is "angular momentum". Therefore I recommend to use this commonly used term, the other terms are rather translation from French.

- Page 10.

** 4th line of the added text, "two finite time" should be "two finite times".

** Eqs (8c) and (8d). I have a problem with the notations. The magnetic moment and the Lagrangian are both scalars or scalar operators. Therefore the two vectors in the right hand side should be separated by dot (scalar product).

- Page 13 (and elsewhere)

* Use of bold characters. I am not sure that it is necessary to use bold for CLF, capital letters are sufficient. Same for CMB and other abbreviations.

- Pages 16-17.

* Figure 3. Subfig 3c can be divided in two subfigures. It is better if the size of these panels can be reduced to hold (together with the caption) in a single page.

- Pages 18-21.

* Same remark for Figures 4 and 6 as for Figure 3.

- Pages 23-27.

A same remark for figures 8 and 9 as for figure 3.

- Pages 31-32.

* Reduce the sizes of subfigures 11 to hold on a single page.

* Section 5. As this section has been largely amended and also because a discussion section was suggested to be added, I recommend to rename this section like "Discussion and conclusions".

* L444. "not need derive" --> does not need to derive

- Page 33.

* L 454. anglular moment--> angular momentum.

* L460. to one another --> to the other

*L492-93. a word is missing in "second derivative of".

Author Response

Please find the list of our corrections and modifications in the uploaded PDF
